# A New Paradigm for Genome-wide DNA Methylation Prediction Without Methylation Input

**Xiaoke Huang**[1*], **Qi Liu**[2*], **Yifei Zhao**[2], **Xianfeng Tang**[4], **Yuyin Zhou**[1†], **Wenpin Hou**[2,3†]
[1] UC Santa Cruz   [2] Columbia University   [3] Duke University   [4] Amazon Research
* indicates equal contribution, † indicates equal advising

## Abstract

DNA methylation (DNAm) is a key epigenetic modification that regulates gene expression and is pivotal in development and disease. However, profiling DNAm at genome scale is challenging: of ~28 million CpG sites in the human genome, only about 1–3% are typically assayed in common datasets due to technological limitations and cost. Recent deep learning approaches, including masking-based generative Transformer models, have shown promise in capturing DNAm–gene expression relationships, but they rely on partially observed DNAm values for unmeasured CpGs and cannot be applied to completely unmeasured samples. To overcome this barrier, we introduce MethylProphet, a gene-guided, context-aware Transformer model for whole-genome DNAm inference without any measured DNAm input. MethylProphet compresses comprehensive gene expression profiles (~25K genes) through an efficient bottleneck multilayer perceptron, and encodes local CpG sequence context with a specialized DNA tokenizer. These representations are integrated by a Transformer encoder to predict site-specific methylation levels. Trained on large-scale pan-tissue whole-genome bisulfite sequencing data from ENCODE (1.6 billion CpG–sample pairs, ~322 billion tokens), MethylProphet demonstrates strong performance in hold-out evaluations, accurately inferring DNAm at unmeasured CpGs and generalizing to unseen samples. Furthermore, application to TCGA pan-cancer data (chromosome 1, 9,194 samples; ~450 million training pairs, 91 billion tokens) highlights its potential for pan-cancer whole-genome methylome imputation. MethylProphet offers a powerful and scalable foundation model for epigenetics, providing high-resolution methylation landscape reconstruction and advancing both biological research and precision medicine.

## 1 Introduction

| | Imputation-based paradigm (e.g., CpGPT, MethylGPT) | Ours |
|---|:---:|:---:|
| CpG-wise ID prediction | ✓ | ✓ |
| CpG-wise OOD prediction w/o fine-tuning | ✗ | ✓ |
| Unseen samples generalization w/o measured DNAm | ✗ | ✓ |
| Multi-omics prediction | ✗ | ✓ |

Table 1: Paradiagm comparison.

DNA methylation (DNAm) is a key epigenetic modification that regulates gene expression, cell differentiation, and disease development (Feinberg, 2018; Loyfer et al., 2023). DNAm predominantly occurs at CpG (cytosine-phosphate-guanine) sites on the DNA sequence, whose tissue-specific and dynamic nature makes them valuable Biomarkers (Hitz et al., 2023; The ENCODE Project Consortium, 2012; Luo et al., 2020; The Cancer Genome Atlas Research Network, 2008). Despite its importance, comprehensive DNAm profiling remains prohibitive. Array-based platforms (e.g., Illumina 450K/EPIC) measure only a small fraction (~1–3%) of the ~28 million CpGs in the human genome, while whole-genome bisulfite sequencing (WGBS) offers complete coverage but at high cost (Shu et al., 2020). As a result, the majority of CpG sites remain unmeasured in typical datasets (Figure 1 (a), Table 1), limiting the insights one can draw from DNAm data.

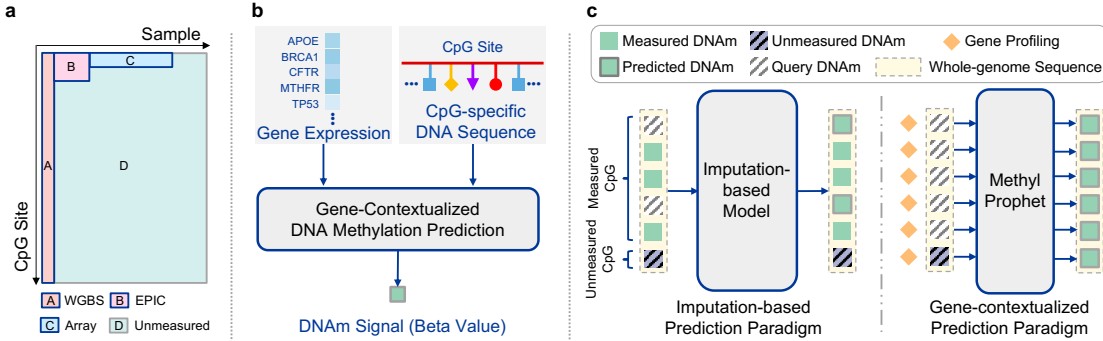

Figure 1: (a) Illustration of the scale of DNAm data. Parts A, B, and C: existing DNAm samples. Part D: unmeasured CpG sites and new samples with gene expression measurements that users can apply MethylProphet to reliably predict their DNAm profiles. (b) Given gene context of a sample, MethylProphet aims to infer whole-genome DNAm at individual CpG resolution. (c) Compared with previous imputation-based methods (*e.g.*, CpGPT (De Lima Camillo et al., 2024)), MethylProphet does not rely on experimentally measured DNAm obtained via wet-lab sequencing, and it directly predicts the target methylation for each CpG site, even those unknown CpG sites.

Table 2: The scale of DNAm data included in this study.

| Data Source | Sequencing Technique | # CpG Sites | # Tissues / Cancers | # Samples | Total #CpG-Sample Pairs | #CpG-Sample Pairs w/ DNAm |
|---|---|---|---|---|---|---|
| ENCODE | WGBS | 27,078,450 | 57 | 95 | 2,572,452,750 | 2,572,452,750 |
| TCGA | Array | 408,399 | 33 | 9,194 | 3,754,820,406 | 3,684,770,086 |
| | EPIC | 740,296 | 4 | 1,706 | 1,262,944,976 | 1,188,102,524 |
| | WGBS | 23,047,052 | 17 | 32 | 737,505,664 | 737,505,664 |

These limitations raise an important question: *Is it feasible to infer DNAm profiles via deep learning models without performing additional sequencing or array experiments?* Gene expression offers a promising source of complementary information, as numerous studies have revealed strong correlations between gene expression levels and DNA methylation patterns (Phillips et al., 2008; Jjingo et al., 2012). Importantly, gene expression data is more readily available across tissues and conditions, and recent advances in large-scale foundation models for genomics suggest that rich biological signals can be learned from such high-dimensional data (Theodoris et al., 2023; Cui et al., 2024a; Yang et al., 2022; Hao et al., 2024a). Motivated by this, we hypothesize that leveraging gene expression can dramatically reduce the reliance on extensive DNAm sequencing, alleviate data scarcity, and enable scalable inference of methylation landscapes.

We propose a novel paradigm for DNAm analysis: using a gene-contextual foundation model to predict a sample's methylome from its gene expression and DNA sequence context alone. This paradigm bypasses the need for any experimentally measured DNAm in the target sample, offering substantial practical advantages. It can lower costs and accelerate analyses by obviating wet-lab experiments, enable studies in resource-limited settings, and potentially reveal DNAm patterns that are otherwise undetectable due to sparse measurements or technical constraints. In line with recent successes in foundation models in genomics (e.g. Geneformer (Theodoris et al., 2023), scGPT (Cui et al., 2024a), scBERT (Yang et al., 2022), and scFoundation (Hao et al., 2024a)), we introduce `MethylProphet` to realize this vision. MethylProphet predicts DNAm by leveraging two key ingredients: **(a)** a **Bottleneck MLP** to compress high-dimensional gene expression profiles (∼25,000 genes) into a compact latent representation, enabling the model to capture global expression patterns and generalize to *unseen samples*; and **(b)** a **DNA sequence tokenizer** to encode local sequence context around each CpG site (e.g., 1 kb window), capturing sequence motifs and epigenetic context that drive methylation at *unseen CpGs*. These gene-derived and sequence-derived embeddings, along with additional genomic annotations (CpG island context, regional genomic features, chromosomal location), are fused by a Transformer encoder (Vaswani, 2017) to output the predicted methylation level for each CpG (Figure 1 (b-c)). By fully leveraging gene expression as context, MethylProphet can infer a

sample's methylome **without requiring any partial DNAm measurements**. This capability stands in contrast to prior methods like DeepCpG (Angermueller et al., 2017), CpGPT (De Lima Camillo et al., 2024), and MethylGPT (Ying et al., 2024), which all depend on some observed DNAm values during inference (imputation paradigm) and thus cannot handle entirely unlabeled samples. Moreover, unlike methods that focus on limited CpG subsets (e.g., MuLan-Methyl (Zeng et al., 2022), StableD-NAm (Zhuo et al., 2023), and MethylNet (Levy et al., 2020)), MethylProphet provides **genome-wide coverage**, overcoming previous coverage limitations.

To rigorously evaluate MethylProphet, we compiled and processed **two billion-scale DNAm datasets** from ENCODE and TCGA (Table 2). The ENCODE dataset consists of whole-genome bisulfite sequencing profiles ($\sim$27 million CpGs) across 95 normal samples (57 tissue types), yielding 1.6 billion CpG–sample training pairs (322 billion input tokens). The TCGA dataset includes 9,194 cancer samples (33 tumor types) with Illumina 450K array data ($\sim$400K CpGs), supplemented by EPIC ($\sim$740K CpGs) and WGBS ($\sim$23 million CpGs in 32 samples) to increase coverage; focusing on chromosome 1, this provides $\sim$450 million training pairs (91 billion tokens). We benchmark MethylProphet in multiple prediction scenarios (with different combinations of unmeasured CpG sites and unseen samples) and observe strong performance—particularly a **median across-sample Pearson correlation (MAS-PCC) for individual CpGs of 0.72** on ENCODE—along with robust accuracy across diverse conditions in TCGA.

Our contributions include:

- **Novel Paradigm**: We develop a flexible and scalable encoding scheme that uniquely integrates gene expression profiles with local DNA sequence context to predict DNAm, without requiring partially measured DNAm as in prior works, overcoming the major limitations of prior imputation-based methods.

- **Scalable Model Design and Benchmarking**: We introduce a modular encoding framework combining an efficient Bottleneck MLP for gene compression with a specialized DNA sequence tokenizer. This design enables scalable training on billions of data points (e.g. 322B tokens from ENCODE, 91B from TCGA) while maintaining tractability and efficiency.

- **Generalization and Practical Impact**: We demonstrate that MethylProphet achieves strong generalization performance across *unmeasured CpGs and unseen samples*. It attains high accuracy (median PCC $\sim 0.7$) on ENCODE and maintains robust performance across various prediction scenarios in TCGA. This foundation model paradigm for methylation inference opens the door to reconstructing complete methylomes from limited experimental data, with broad implications for epigenetic research and precision medicine[1].

## 2 BACKGROUND AND RELATED WORKS

**DNAm Data Scale and Coverage.** DNA methylation (DNAm) can be represented as a CpG-by-sample matrix $M \in \mathbb{R}^{N_{CpG} \times N_s}$, where $N_{CpG} \approx 2.8 \times 10^7$ sites genome-wide and the entries are methylation levels ($\beta$ values $\in [0, 1]$).. Existing assays trade off coverage and cost: array-based platforms capture only 1–3% of CpGs ($\approx 10^5$ sites) (Shu et al., 2020), while WGBS provides nearly complete coverage but remains costly for large cohorts. This disparity means that most CpGs are typically unmeasured in any given dataset, creating a high-dimensional and massive missing data problem. A naive approach of assigning each CpG site a unique learnable parameter (embedding) in a model would be infeasible: it require $\sim$86GB for all $2.8 \times 10^7$ CpGs with 768-dimentional embeddings, and it fails to generalize to unseen CpGs (Figure 1(a), Table 2).

**Gene Expression as Context.** Gene expression can be represented as a Gene-by-sample matrix $G \in \mathbb{R}^{N_g \times N_s}$, where $N_g \approx 20000$ makes direct Transformer encoding intractable due to quadratic complexity of self-attention. However, capturing the full gene expression landscape is crucial, as it provides global biological state that can inform local methylation states.

These characteristics demand novel architectural solutions that can effectively (1) *represent and generalize across millions of CpG sites*, and (2) *efficiently process comprehensive gene expression profiles while maintaining computational tractability*.

---

[1]Detailed insights on the biological implications are provided in the Appendix.

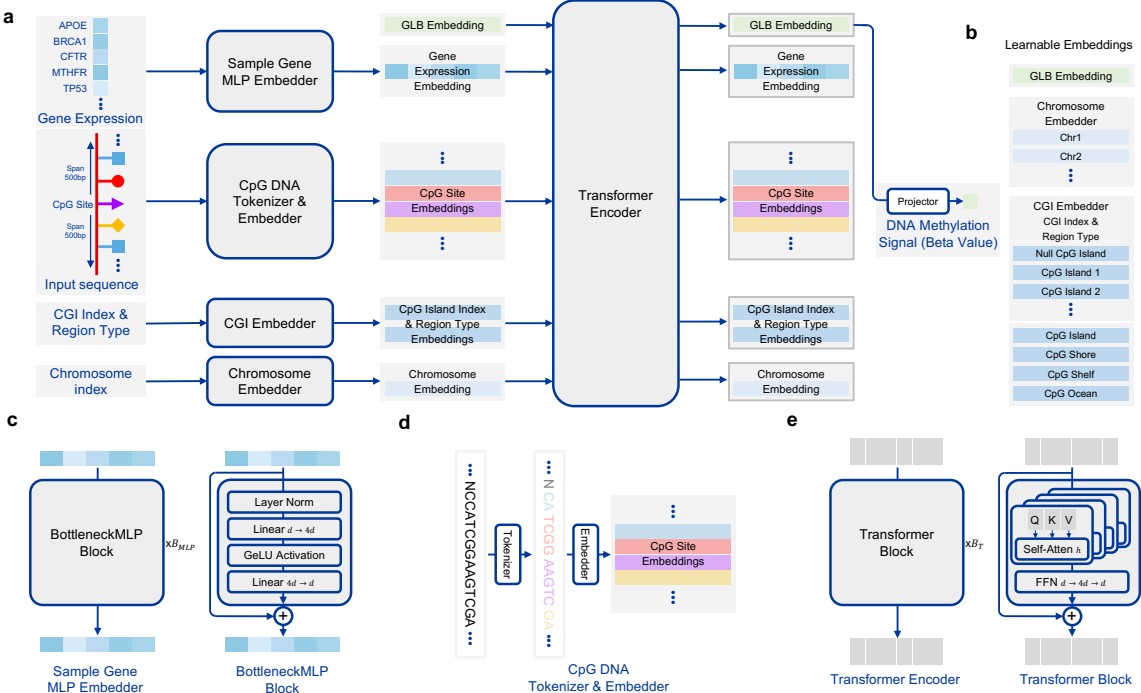

Figure 2: **Overview of our proposed pipeline.** (a) Model architecture of MethylProphet; (b) The learnable Global, chromosome, and CPG island-related embeddings; (c) Model architecture of efficient gene profile compression MLP; (d) DNA Tokenizer for CpG-specific DNA sequence; (e) Model architecture of the Transformer encoder that aggregates all the embeddings.

Prior computational methods for DNAm imputation or prediction have been limited in scope. Traditional approaches and early deep learning models (Zeng et al., 2022; Angermueller et al., 2017; Zhuo et al., 2023; Levy et al., 2020; Wang et al., 2024; Levy-Jurgenson et al., 2019a) targeted DNAm prediction at a limited subset of CpG sites (e.g. those on arrays or select regions) and/or specific sample sets. More recent Transformer-based models (e.g. CpGPT (De Lima Camillo et al., 2024), MethylGPT (Ying et al., 2024)) leveraged the power of attention mechanisms, but they employed masked modeling pre-training on the order of $10^4$ CpGs ($\sim$0.03% of the genome) to learn a latent representation of the methylome. Because their objective is to learn holistic representations for downstream tasks rather than directly predict missing values, these models still require some observed methylation input and do not generalize to completely unseen CpG sites or new samples. Furthermore, many existing methods do not integrate critical biological context such as gene expression, or they face scaling bottlenecks when attempting to handle genome-wide patterns. Notably, Levy-Jurgenson et al. (2019a) proposed a deep model with attention to predict methylation from gene expression and sequence, but it was only demonstrated on thousands of CpGs, a limited sample size, and cancer cohorts.

In contrast, our work offers a comprehensive gene-contextual solution that addresses these gaps. MethylProphet distinguishes itself by operating at full methylome scale and by leveraging a foundation-model approach: it is trained on billions of CpG–sample pairs to directly infer complete methylation profiles, enabling generalization to unmeasured CpGs and unseen samples in a way that prior methods could not achieve.

## 3    METHYLPROPHET MODEL

MethylProphet is a gene-contextual Transformer capable of learning the whole-genome DNAm landscape by integrating genome-wide gene expression with CpG-specific DNA sequence context. The model architecture (Figure 2) consists of distinct modules for encoding the sample's gene expression profile and the target CpG site's context, which are combined within a Transformer to produce a

methylation prediction. We adopt a Transformer encoder (Vaswani, 2017) because its self-attention natively captures the long-range dependencies that link distant CpG sites within kilobase-scale DNA sequences (De Lima Camillo et al., 2024), seamlessly fuses heterogeneous embeddings of sequence, gene expression, and genomic annotations without bespoke cross-modality modules (Gao et al., 2024; Cui et al., 2024a; Yang et al., 2022). In addition, it exhibits a well-established scaling law (Kaplan et al., 2020; Henighan et al., 2020), *i.e.*, more data consistently translate to better performance, making it an ideal backbone for whole-genome DNAm prediction.

**Problem Formulation.** Let $\mathcal{G} \in \mathbb{R}^{N_g}$ denote the expression vector of $N_g \approx 25000$ genes for a given sample, and let $S_i \in \{A, T, C, G\}^L$ be the DNA sequence of length $L$ centered on CpG site $i$. Each CpG has auxiliary annotations $a_i$ (e.g., CpG island index, genomic region, chromosome). Our goal is to learn a function

$$f_\theta : (\mathcal{G}, S_i, a_i) \mapsto \hat{y}_i \in [0, 1], \tag{1}$$

where $\hat{y}_i$ is the predicted DNAm level of CpG $i$, and $y_i$ is the ground-truth DNAm from sequencing.

**Gene Expression Bottleneck MLP (Figure 2 (a, c))**: We employ a bottleneck MLP (Bachmann et al., 2023) that compresses high-dimensional gene expression profile $\mathcal{G}$ into a compact latent embedding $x_{\text{gene}} \in \mathbb{R}^{N_{\text{embed}}}$: $x_{\text{gene}} = \phi(W_2\,\sigma(W_1\mathcal{G} + b_1) + b_2)$, where $W_1 \in \mathbb{R}^{N_h \times N_g}$, $W_2 \in \mathbb{R}^{N_{\text{embed}} \times N_h}$, $\sigma$ is the GeLU activation, and $\phi$ is layer normalization. Unlike approaches that attempted to attend only thousands of gene tokens (Cui et al., 2024b; Bai et al., 2024; Hao et al., 2024b), this design (i) compresses $\sim 25000$ genes efficiently, (ii) introduces minimal inductive bias, (iii) preserves long-range dependencies across the transcriptome, and (iv) generalizes to unseen samples.

**CpG Sequence Tokenizer & Context Embeddings (Figure 2 (a, d))**: To represent each target CpG site in a way that generalizes across millions of possible loci, we do not assign a fixed ID. Instead, we encode a CpG by its local genomic sequence context. We utilize a DNA sequence tokenizer inspired by DNABERT-2 (Zhou et al., 2024), which applies a variable-length byte-pair encoding (BPE) scheme to the DNA sequence surrounding the CpG. Specifically, for each CpG site we take a window of e.g. 1000 base pairs (bp) centered on the site. This sequence $S$ (length 1kb, consisting of characters A,T,C,G) is broken into a sequence of subword tokens $T = t_j$ via the DNA tokenizer (Figure 2 (a,d)). The tokenizer compresses repetitive or common motifs, achieving roughly a 5× reduction in length (1,000bp $\rightarrow \sim$200 tokens) while preserving biologically relevant patterns. Each token $t_j$ is then mapped to a learnable embedding vector $x_j^{\text{DNA}} \in \mathbb{R}^{N_{\text{embed}}}$. This tokenization approach has several benefits: it identifies and reuses recurring sequence motifs (e.g., CpG-rich patterns or regulatory motifs), reduces redundancy, and yields a consistent embedding length for any CpG's context. Importantly, similar sequence patterns will produce similar token sequences, allowing the model to generalize knowledge across different CpG sites that share motifs. In addition to raw sequence, we incorporate genomic context features that help distinguish CpG sites:

- **CpG island (CGI) context (Figure 2 (a, b)**: DNAm behavior differs if a CpG lies within a CpG island, shore, shelf, or open sea (ocean). We include a CpG island index embedding to provide a unique identifier for each CpG island (with a special index for non-island CpGs in open sea), as well as separate embeddings for region categories (island, shore, shelf, ocean). By summing the island-specific embedding with the region-type embedding, we obtain a composite context vector $x_{\text{CGI}}$ for the site. This encoding injects knowledge of local CpG density and regulatory regions, helping the model resolve ambiguity when similar DNA sequences appear in different contexts.

- **Chromosome indicator (Figure 2 (b))**: We assign each chromosome a learnable embedding $x_{\text{chr}(k)} \in \mathbb{R}^{N_{\text{embed}}}$ for chromosome $k$ ($k = 1, \ldots, 22$). This provides positional information that can capture chromosome-specific effects (such as varying methylation baseline or sequence composition) and helps the model differentiate sites that may have similar sequence but belong to different genomic compartments.

- **Global and integrated representations (Figure 2 (a))**: Following conventions in Transformer models (Devlin, 2018) and prior genome foundation models(Cui et al., 2024b; Bai et al., 2024; Hao et al., 2024b), we apply a learnable global embedding token $x_{\text{GLB}}$. This vector does not correspond to any specific gene or CpG input; instead, it serves as an aggregate representation that can attend to all other embeddings. The Transformer can use $x_{\text{GLB}}$ to gather information across the gene expression context and the target CpG context. At the output, this global token's state will be fed into a prediction head to produce the final methylation level for the CpG site in the given sample.

**Transformer Encoder and Prediction (Figure 2 (a))**: We concatenate all embedding vectors for a given sample–CpG pair into a single sequence:

$$Z_i = \left[ x_{\text{GLB}}, \ x_{\text{gene}}, \ \{x_j^{\text{DNA}}\}_{j=1}^L, \ x_{\text{CGI}}, \ x_{\text{chr}} \right], \tag{2}$$

where $x_{\text{gene}}$ is the gene expression embedding, $x_j^{\text{DNA}}$ are the DNA sequence token embeddings, $x_{\text{CGI}}$ encodes CpG island status, $x_{\text{chr}}$ encodes the chromosome, and $L$ is the DNA sequence length (default $L = 2000$). This sequence $Z_i$ forms the input to the Transformer encoder. It consists of stacked self-attention layers that fuse information across modalities. The bi-directional attention mechanism allows each token to attend to every other token in the sequence, enabling the model to combine global sample context with local site context effectively. For example, the gene expression embedding can influence how sequence tokens are interpreted and vice versa, allowing complex interactions (e.g. gene regulatory network effects on local methylation) to emerge in the learned representation. The Transformer encoder outputs contextualized embeddings of the same length. We then apply a simple DNAm projector: a linear layer followed by a sigmoid activation, acting on the final state of the global token $x_{\text{GLB}}^{\text{out}}$, to predict the DNAm for the target CpG in the sample. This design, using a global token as the prediction carrier, is analogous to the "CLS" token in BERT models and encourages the network to integrate all information into that token for the final regression output.

**Training Objective**: MethylProphet is trained end-to-end to minimize mean squared error (MSE) between predicted and true methylation values, with all components (gene MLP, tokenizer embeddings, Transformer, etc.) updated via back-propagation. Training is fully supervised on large gene expression–DNAm datasets, while inference requires only gene expression and sequence data. See Appendix for implementation and training details.

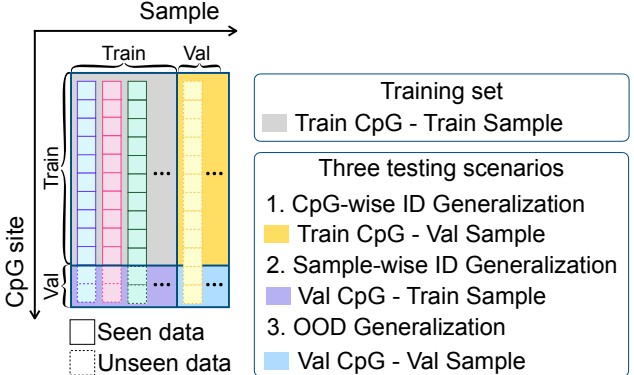

Figure 3: MethylProphet can predict unseen CpGs and unseen samples without inputting measured DNAm levels. Each column is the genome-wide CpG-level DNAm of a sample.

## 4 EXPERIMENTS

### 4.1 DATA SOURCE, PROTOCOLS, AND PRE-PROCESSING

We use ENCODE (Hitz et al., 2023; The ENCODE Project Consortium, 2012; Luo et al., 2020) and TCGA data (The Cancer Genome Atlas Research Network, 2008) for model evaluation. The ENCODE data contains 97 pairs of matched WGBS and RNA-seq samples across normal tissue types and cell types. We use it for evaluating MethylProphet's performance on predicting pan-tissue whole-genome DNAm. The TCGA data contains 10,932 pairs of matched DNAm (including WGBS (W), EPIC 850K (E), and 450K array (A)) and RNA-seq samples. We use it for evaluating MethylProphet's performance on pan-cancer prediction. Together, they provide billions of DNA methylation signals at individual CpGs with matched gene expression profiles. To evaluate in- and out-of-distribution generalization, we partitioned both samples and CpG sites, with train CpG–train sample for training and the other three splits for validation (Figure 3, Table 3). See Appendix for data and preprocessing steps.

### 4.2 BASELINE COMPARISONS

We evaluated MethylProphet's performance on all three validate splits (Figure 3). To quantify agreement between predicted and measured DNAm values, we employ the following metrics (more analysis is provided in Appendix): (a) *Median-across-CpG Pearson correlation coefficient (MAC-PCC)* assesses how well the model preserves each sample's overall DNAm profile. (b) *Median-across-sample PCC (MAS-PCC)* evaluates the model's ability to infer DNAm behavior at individual CpG sites. (c) *Mean square error (MSE)* evaluates the difference between predicted and measured DNAm. (d) *Mean absolute error (MAE)* evaluates the absolute difference between predicted and measured DNAm.

Table 3: The data statistics among all the data source and splits in our experiments. The number of tokens is estimated by the average sequence length (*i.e.*, 200) of the input embeddings of the Transformer encoder.

| Dataset | Chr. | Sequen. | Split | # CpG Sites | # Tissues | # Samples | # Pairs w/ Me. | # Tokens |
|---|---|---|---|---|---|---|---|---|
| ENCODE | 1 - 22 | WGBS | Train: Train CpG - Train Sample | 24, 363, 170 | 57 | 66 | 1, 607, 969, 220 | 321, 593, 844, 000 |
| | | | Val: Train CpG - Val Sample | 24, 363, 170 | 22 | 29 | 706, 531, 930 | 141, 306, 386, 000 |
| | | | Val: Val CpG - Train Sample | 2, 707, 033 | 57 | 66 | 178, 664, 178 | 35, 732, 835, 600 |
| | | | Val: Val CpG - Val Sample | 2, 707, 033 | 22 | 29 | 78, 503, 957 | 15, 700, 791, 400 |
| TCGA | 1 | Array | Train: Train CpG - Train Sample | 33, 885 | 33 | 8, 258 | 275, 018, 849 | 55, 003, 769, 800 |
| | | EPIC | | 71, 748 | 4 | 1, 706 | 115, 856, 100 | 23, 171, 220, 000 |
| | | WGBS | | 1, 999, 446 | 17 | 32 | 63, 982, 272 | 12, 796, 454, 400 |
| | | Array | Val: Train CpG - Val Sample | 33, 885 | 33 | 920 | 30, 638, 464 | 6, 127, 692, 800 |
| | | | Val: Val CpG - Train Sample | 6, 742 | 33 | 8, 258 | 55, 141, 308 | 11, 028, 261, 600 |
| | | | Val: Val CpG - Val Sample | 6, 742 | 33 | 920 | 6, 143, 360 | 1, 228, 672, 000 |
| TCGA | 1 - 3 | Array | Train: Train CpG - Train Sample | 78, 211 | 33 | 8, 258 | 632, 281, 133 | 126, 456, 226, 600 |
| | | EPIC | | 172, 722 | 4 | 1, 706 | 276, 181, 739 | 55, 236, 347, 800 |
| | | WGBS | | 5, 396, 193 | 17 | 32 | 172, 678, 176 | 34, 535, 635, 200 |
| | | Array | Val: Train CpG - Val Sample | 78, 211 | 33 | 920 | 70, 443, 801 | 14, 088, 760, 200 |
| | | | Val: Val CpG - Train Sample | 14, 893 | 33 | 8, 258 | 121, 617, 682 | 24, 323, 536, 400 |
| | | | Val: Val CpG - Val Sample | 14, 893 | 33 | 920 | 13, 550, 097 | 2, 710, 019, 400 |

We applied all metrics when benchmarking MethylProphet'performance. Metrics (a)-(d) were applied in the comparisons with CNN-based attention model (Levy-Jurgenson et al., 2019b). MethylProphet consistently outperforms this baseline across all settings (Table 4, 5).

Table 4: Performance comparison on ENCODE data. Four evaluation metrics are included: median-across-sample PCC (MAS-PCC), median-across-sample PCC (MAS-PCC), mean square error (MSE), and mean absolute error (MAE).

| | Train CpG - Val Sample | | | | Val CpG - Train Sample | | | | Val CpG - Val Sample | | | |
|---|---|---|---|---|---|---|---|---|---|---|---|---|
| | MAS-PCC | MAC-PCC | MSE | MAE | MAS-PCC | MAC-PCC | MSE | MAE | MAS-PCC | MAC-PCC | MSE | MAE |
| Levy-Jurgenson et al. (2019b) | 0.2878 | 0.8355 | 0.0182 | 0.0875 | 0.5453 | 0.7959 | 0.0345 | 0.1250 | 0.1930 | 0.8037 | 0.0343 | 0.1262 |
| MethylProphet | 0.3436 | 0.9398 | 0.0079 | 0.0608 | 0.7165 | 0.9297 | 0.0108 | 0.0679 | 0.3411 | 0.9330 | 0.0086 | 0.0634 |

Table 5: Performance comparison on TCGA data.

| | Train CpG - Val Sample | | | | Val CpG - Val Sample | | | | Val CpG - Val Sample | | | |
|---|---|---|---|---|---|---|---|---|---|---|---|---|
| | MAS-PCC | MAC-PCC | MSE | MAE | MAS-PCC | MAC-PCC | MSE | MAE | MAS-PCC | MAC-PCC | MSE | MAE |
| Sample mean baseline | N/A | 0.9686 | 0.0048 | 0.0471 | N/A | N/A | N/A | N/A | N/A | N/A | N/A | N/A |
| Levy-Jurgenson et al. (2019b) | 0.2630 | 0.6325 | 0.0874 | 0.2148 | 0.2203 | 0.6563 | 0.0848 | 0.2048 | 0.2158 | 0.6562 | 0.0854 | 0.2055 |
| MethylProphet | 0.5455 | 0.9320 | 0.0199 | 0.0882 | 0.4194 | 0.9065 | 0.0266 | 0.1000 | 0.3904 | 0.9059 | 0.0271 | 0.1011 |

In addition, although CpGPT and MethylGPT rely on a different paradigm and either do not predict for unseen samples, we evaluated their in-distribution generalization. MethylProphet achieves the highest MAS-PCC and MAC-PCC (Table 6).

Table 6: Comparisons on "Val CpG - Train Sample" split (in-distribution inference) across models.

| | ENCODE | | TCGA chr1 | |
|---|---|---|---|---|
| Model | MAS-PCC | MAC-PCC | MAS-PCC | MAC-PCC |
| DeepCPG | 0.0317 | 0.5560 | -0.0080 | 0.4237 |
| CpGPT | 0.3192 | 0.9401 | 0.4794 | 0.9250 |
| MethylGPT | 0.2964 | 0.8953 | 0.4358 | 0.8357 |
| **MethylProphet** | **0.3689** | **0.9400** | **0.5453** | **0.9253** |

## 4.3 ANALYSIS PROCEDURE

We apply MethylProphet to predict DNAm in the TCGA and ENCODE datasets, cross-validate its predictions, and analyze their biological implications. Specifically, *CGI coherence* is evaluated by measuring correlation between CpG pairs within the same CGI. As a baseline, we compute correlations after randomly permuting CGI indicators. *Uniform Manifold Approximation and Projection*

*for Dimension Reduction (UMAP)* (Leland McInnes, 2018) verifies whether the inferred DNAm landscape captures tissue and cancer differences while preserving variation.

**MethylProphet performance on ENCODE data**   The across-CpG PCC (Figure 4(a, d)) reaches highest in the *Train CpG - Val Sample* split, indicating that the model effectively captures site-wise DNAm patterns while generalizing well to new samples. If a sample exhibits high across-CpG PCC, it suggests that the within-sample variability of CpGs is well captured (Figure 4 (d)). This result is expected, as the overall DNAm profile of a sample consists of a long vector of CpG elements, and global trends in DNAm are typically easier to learn and predict. For across-sample PCC (Figure 4 (b)), we observe a large variability, particularly when generalizing to unmeasured CpGs and unseen samples. The CpGs with high across-sample PCC indicate that the model can predict the CpG's variability across samples (Figure 4 (e)) well. This is very important because the ability to predict a CpG's behavior across individuals is highly related to its potential to identify a therapeutic target.

Specifically, the predictions are not only accurate when generalizing to unmeasured CpGs which is a task that existing methods such as CpGPT can do, but MethylProphet also achives satisfactory performance in unmeasured samples (samples not seen at all for the model, and samples without experimentally-measured DNAm) which existing methods cannot do (Figure 4 (d, e)). For across-CpG PCC (Figure 4 (a, d)), the performance is similar across splits, while for across-sample PCC (Figure 4(b, e)), MethylProphet performs best in the *Val CpG - Train Sample* split, possibly due to the limited testing samples in ENCODE data. Further investigation show that the predictions are more accurately for highly variable CpGs, where across-sample PCC increases with CpG variability. In this normal tissue cohort, MethylProphet also effectively captures CpG co-methylation dynamics within CGIs (Figure 4 (c)). In addition, MethylProphet performs comparably across splits, likely due to the significantly large number of CpGs. Overall, MethylProphet successfully preserves tissue differences (Figure 4 (f)), with predicted and measured samples of the same tissue types clustering together.

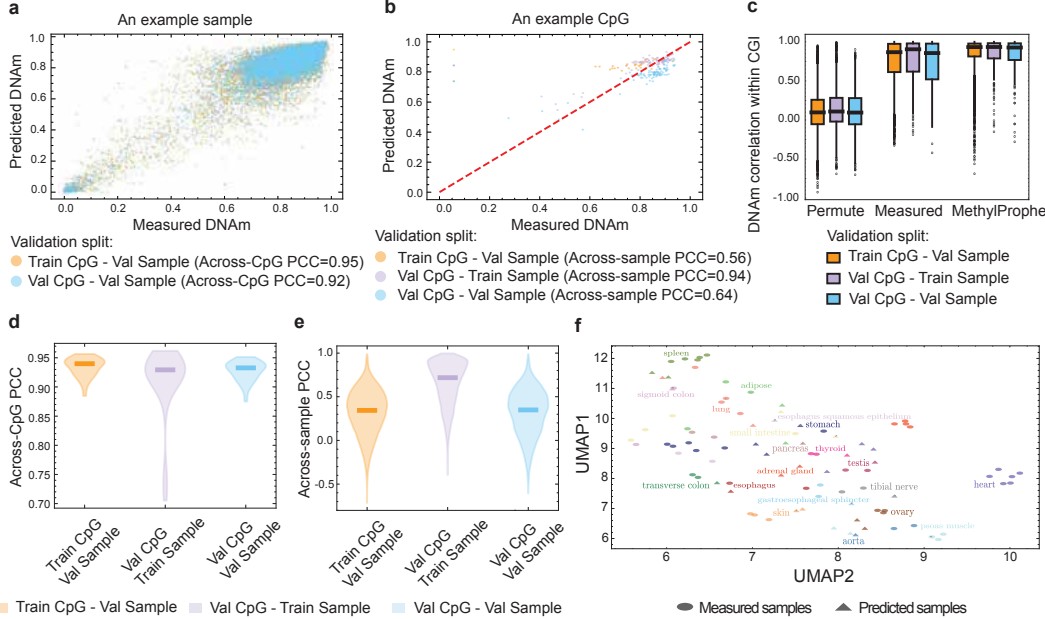

Figure 4: Cross-validation results on ENCODE data. (a) An example sample to demonstrate the calculation of across-CpG PCC. (b) An example CpG to demonstrate the calculation of across-sample PCC. (c) Predicted signal similarity within CGIs. (d) Across-CpG PCC. (e) Across-sample PCC. (f) UMAP of measured (triangles) and predicted (circles) samples.

**MethylProphet performance on TCGA data**   Evalution results demonstrated MethylProphet's strong pan-cancer prediction performance (see Appendix).

## 4.4 Ablation Studies

**Data mixing strategies and scaling effects** In the data mixing ablation (Table 7), MethylProphet demonstrates strong CpG encoding capability on ENCODE WGBS data, achieving high MAS-PCC scores of 0.72 on the Val CpG - Train Sample split. However, performance on Val Sample is moderate, likely due to the limited sample size (Table 3) constraining the model's ability to learn generalized gene encodings for sample-specific features. Training solely on Array data yields suboptimal performance, with MAS-PCC scores of 0.40, 0.28, and 0.26 on Train CpG - Val Sample, Val CpG - Train Sample, and Val CpG - Val Sample splits, respectively. This limitation stems from the restricted number of CpG sites in Array data (Table 3). Performance improves consistently when incorporating additional data sources, with optimal results achieved by combining Array, EPIC, and WGBS data, yielding MAS-PCC scores of 0.54, 0.42, and 0.39 for the respective splits.

Table 7: Results of training models on different data sources. Datasets: E and T denotes ENCODE and TCGA, respectively; in brackets, A, E, and W denote Array, EPIC, and WGBS samples, respectively.

| Train Data | Val Data | Train CpG - Val Sample | | | | Val CpG - Train Sample | | | | Val CpG - Val Sample | | | |
|---|---|---|---|---|---|---|---|---|---|---|---|---|---|
| | | MAS-PCC | MAC-PCC | MSE | MAE | MAS-PCC | MAC-PCC | MSE | MAE | MAS-PCC | MAC-PCC | MSE | MAE |
| E (W) | E (W) | 0.3436 | 0.9398 | 0.0079 | 0.0608 | 0.7165 | 0.9297 | 0.0108 | 0.0679 | 0.3411 | 0.9330 | 0.0086 | 0.0634 |
| T (A) | T (A) | 0.4000 | 0.8669 | 0.0363 | 0.1216 | 0.2769 | 0.7914 | 0.0555 | 0.1498 | 0.2597 | 0.7930 | 0.0557 | 0.1504 |
| T (A+W) | T (A) | 0.4705 | 0.3244 | 0.2981 | 0.9112 | 0.8674 | 0.8673 | 0.0252 | 0.0365 | 0.0369 | 0.1006 | 0.1205 | 0.1212 |
| T (A+E) | T (A) | 0.5226 | 0.9232 | 0.0222 | 0.0920 | 0.3727 | 0.8738 | 0.0350 | 0.1147 | 0.3451 | 0.8743 | 0.0355 | 0.1157 |
| T (A+E+W) | T (A) | 0.5455 | 0.9320 | 0.0199 | 0.0882 | 0.4194 | 0.9065 | 0.0266 | 0.1000 | 0.3904 | 0.9059 | 0.0271 | 0.1011 |

The scaling ablation (Table 8) reveals that models trained exclusively on chromosome 1 show limited generalization to additional chromosomes (2 and 3). While training on all three chromosomes slightly decreases validation performance on chromosome 1, it significantly improves the model's ability to generalize across chromosomes.

Table 8: The ablation of increasing training data scale by adding chromosomes for TCGA data.

| Train Chr. | Val Chr. | Train CpG - Val Sample | | | | Val CpG - Train Sample | | | | Val CpG - Val Sample | | | |
|---|---|---|---|---|---|---|---|---|---|---|---|---|---|
| | | MAS-PCC | MAC-PCC | MSE | MAE | MAS-PCC | MAC-PCC | MSE | MAE | MAS-PCC | MAC-PCC | MSE | MAE |
| 1 | 1 | 0.5455 | 0.9320 | 0.0199 | 0.0882 | 0.4194 | 0.9065 | 0.0266 | 0.1000 | 0.3904 | 0.9059 | 0.0271 | 0.1011 |
| 1+2+3 | 1 | 0.4928 | 0.9249 | 0.0219 | 0.0915 | 0.3760 | 0.8961 | 0.0294 | 0.1047 | 0.3505 | 0.8960 | 0.0298 | 0.1057 |
| 1 | 1+2+3 | 0.3025 | 0.8012 | 0.0535 | 0.1473 | 0.2654 | 0.8216 | 0.0492 | 0.1362 | 0.2513 | 0.8230 | 0.0495 | 0.1368 |
| 1+2+3 | 1+2+3 | 0.4872 | 0.9246 | 0.0224 | 0.0919 | 0.3736 | 0.8993 | 0.0290 | 0.1027 | 0.3460 | 0.8992 | 0.0295 | 0.1037 |

**Contribution of gene encoding** Table 9 compares MethylProphet with alternative gene encoding strategies, as well as a variant without gene encoding, on TCGA data. In the gene-pathway encoder, the Bottleneck MLP is replaced with a sparse MLP (Jaume et al., 2024). Across all evaluation metrics, MethylProphet consistently achieves the best performance. See Appendix for implementation details.

Table 9: The ablation of different gene encoding strategies for TCGA data.

| Cases | Train CpG - Val Sample | | | | Val CpG - Val Sample | | | | Val CpG - Val Sample | | | |
|---|---|---|---|---|---|---|---|---|---|---|---|---|
| | MAS-PCC | MAC-PCC | MSE | MAE | MAS-PCC | MAC-PCC | MSE | MAE | MAS-PCC | MAC-PCC | MSE | MAE |
| gene-pathway encoder | 0.5371 | 0.9256 | 0.0212 | 0.0907 | **0.4194** | 0.9043 | 0.0271 | 0.1012 | **0.3959** | 0.9018 | 0.0280 | 0.1029 |
| DNA-seq encoder only | N/A | 0.8539 | 0.0413 | 0.1393 | N/A | 0.8607 | 0.0400 | 0.1304 | N/A | 0.8607 | 0.0404 | 0.1311 |
| MethylProphet | **0.5455** | **0.9320** | **0.0199** | **0.0882** | **0.4194** | **0.9065** | **0.0266** | **0.1000** | 0.3904 | **0.9059** | **0.0271** | **0.1011** |

**Contribution of genomic annotations** Table 10 compares MethylProphet with and without genomic annotations (i.e., CpG island (CGI) annotation and chromosome identity).

## 4.5 Computational Efficiency and Practical Deployment

MethylProphet is designed for practical deployment, requiring modest GPU memory and runtime. Table 11 shows benchmarks on NVIDIA L40s GPU (48GB), confirming that real-world applications are feasible without heavy hardware demands.

Table 10: The ablation of genomic annotations (i.e., CGI and chromosome) for MethylProphet. Datasets: E and T denotes ENCODE and TCGA, respectively; in brackets, A, E, W and digit denote Array, EPIC, and WGBS samples, chromosome No., respectively.

| Train Data | Val Data | Model | Train CpG - Val Sample | | | | Val CpG - Train Sample | | | | Val CpG - Val Sample | | | |
|---|---|---|---|---|---|---|---|---|---|---|---|---|---|---|
| | | | MAS-PCC | MAC-PCC | MSE | MAE | MAS-PCC | MAC-PCC | MSE | MAE | MAS-PCC | MAC-PCC | MSE | MAE |
| E (W) | E (W) | MethylProphet w/o CGI | 0.3357 | 0.9208 | 0.0083 | 0.0635 | 0.6776 | 0.9196 | 0.0133 | 0.0692 | 0.3251 | 0.9222 | 0.0090 | 0.0645 |
| E (W) | E (W) | MethylProphet w/o chromosome | 0.3105 | 0.9216 | 0.0083 | 0.0685 | 0.6959 | 0.9123 | 0.0117 | 0.0681 | 0.3328 | 0.9202 | 0.0090 | 0.0690 |
| E (W) | E (W) | MethylProphet | 0.3436 | 0.9398 | 0.0079 | 0.0608 | 0.7165 | 0.9297 | 0.0108 | 0.0679 | 0.3411 | 0.9330 | 0.0086 | 0.0634 |
| T (A+E+W,1) | T (A,1) | MethylProphet w/o CGI | 0.5353 | 0.9238 | 0.0221 | 0.0896 | 0.3948 | 0.8965 | 0.0285 | 0.1460 | 0.3782 | 0.8948 | 0.0300 | 0.1292 |
| T (A+E+W,1) | T (A,1) | MethylProphet | 0.5455 | 0.9320 | 0.0199 | 0.0882 | 0.4194 | 0.9065 | 0.0266 | 0.1000 | 0.3904 | 0.9059 | 0.0271 | 0.1011 |
| T (A+E+W,1+2+3) | T (A,1+2+3) | MethylProphet w/o chromosome | 0.4631 | 0.9159 | 0.0272 | 0.1001 | 0.3595 | 0.8709 | 0.0354 | 0.1264 | 0.3389 | 0.8852 | 0.0350 | 0.1215 |
| T (A+E+W,1+2+3) | T (A,1+2+3) | MethylProphet | 0.4872 | 0.9246 | 0.0224 | 0.0919 | 0.3736 | 0.8993 | 0.0290 | 0.1027 | 0.3460 | 0.8992 | 0.0295 | 0.1037 |

Table 11: Inference benchmarks of MethylProphet on L40s GPU (48GB). Each benchmark uses $2.7 \times 10^7$ CpG sites across 10 samples.

| Number of Samples | Number of CpGs | Time | GPUs | Batch size | Memory |
|---|---|---|---|---|---|
| 10 | $2.7 \times 10^7$ | $\sim$9 min | 32 | 256 | 12.3 GB |
| 10 | $2.7 \times 10^7$ | $\sim$17 min | 16 | 256 | 12.3 GB |
| 10 | $2.7 \times 10^7$ | $\sim$34 min | 8 | 256 | 12.3 GB |
| 10 | $2.7 \times 10^7$ | $\sim$68 min | 4 | 256 | 12.3 GB |

## 5 CONCLUSION

We present MethylProphet, a novel Transformer-based approach that enables whole-genome DNA methylation inference by integrating gene profile with genomic context. Trained on extensive datasets, our model demonstrates robust performance in inferring genome-wide methylation patterns across diverse tissues and cancer types. We hope this capability to reconstruct complete methylomes from limited experimental data could advance both epigenetic research and precision medicine applications.

## ACKNOWLEDGEMENTS

W. H., Q. L, and Y.F. Z. were supported by the National Human Genome Research Institute of the National Institutes of Health under Award Number K99/R00HG011468, through which W. H. received related training from Drs. Hongkai Ji, Stephanie Hicks, and Andrew Feinberg. Q. L. was also partially supported by the General Fund at Columbia University Department of Biostatistics. We would like to thank Columbia HPC (including Insomnia and C2B2), TPU Research Cloud (TRC) program and Google Cloud Research Credits program for supporting our computing needs. This research was partially supported by the UC National Laboratory Fees Research Program of the University of California Office of the President (Grant L26CR10102).

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

# Appendix for "MethylProphet"

## Contents

## A  POTENTIAL BIOLOGICAL INSIGHTS

METHYLPROPHET enables genome-wide DNA methylation (DNAm) reconstruction from gene expression and sequence data alone, providing unique opportunities for advancing biological interpretation, methodological development, and genomic applications. This cross-modality prediction framework offers several important insights and use cases in real-world biomedical research.

First, METHYLPROPHET facilitates low-cost methylome reconstruction in settings where whole-genome bisulfite sequencing (WGBS) or array-based profiling is infeasible. Many large-scale transcriptomic datasets lack matching methylome profiles, and thus cannot be directly leveraged for epigenetic discovery. For example, the ENCODE consortium has generated 1,699 RNA-seq samples but only 211 WGBS samples; the TCGA program includes more than 10,426 RNA-seq samples but only 32 WGBS samples; and GEO hosts 241,014 RNA-seq samples but just 6,318 WGBS samples. By computationally inferring DNAm in these cohorts, METHYLPROPHET enables downstream epigenetic analyses without the need for additional profiling.

Second, METHYLPROPHET enhances public and disease biobank resources such as GTEx, ENCODE, TCGA, and PCAWG by providing whole-genome methylome predictions. This allows for deeper epigenetic insights, cancer subtype stratification, and biomarker discovery. Prior work, such as Yang et al. (2024), predicted DNAm from GTEx and multi-omics TCGA data, but their scope was limited to Illumina EPIC array CpGs, covering only ∼3% of the genome. By contrast, METHYLPROPHET enables whole-genome prediction at more than $100\times$ the sample scale, thereby extending coverage from 3% to 100% of the genome and broadening the landscape of epigenetic discovery.

Third, METHYLPROPHET supports sample-level methylation estimation in multi-omic and single-cell studies, where DNAm data are often sparse or missing. This ability to reconstruct complete sample-level methylomes from transcriptomic profiles enables downstream tasks such as DNAm regulation inference, cell-fate trajectory analysis, and multi-omic clustering, all without requiring methylation-specific assays.

In addition, METHYLPROPHET contributes to predictive biomarker development. For instance, 850K array-based methylation profiles have been used to predict brain metastases (Zuccato et al., 2025). By extending methylation reconstruction to the full genome, METHYLPROPHET opens new possibilities for noninvasive biomarker discovery and risk stratification in cohorts that lack direct methylation assays.

Another important application is in the development of DNA methylation clocks for aging and disease phenotyping. Epigenetic clocks such as Horvath and GrimAge estimate biological age based on a small number of CpGs, but their accuracy is limited by array coverage (1–3% of the genome). METHYLPROPHET provides genome-wide methylation inference, improving both the resolution and accuracy of aging models. Furthermore, it enables biological age estimation in transcriptome-only cohorts, thereby expanding the reach of age-related biomarkers in large-scale population and longitudinal studies.

Beyond these applications, METHYLPROPHET establishes cross-modality prediction as a powerful paradigm in multi-omics. Cross-modal inference is increasingly central to computational biology: studies have predicted DNAm from expression (Yang et al., 2024; Liu et al., 2024), chromatin accessibility from expression and DNA (Zhou et al., 2017), and gene expression from sequence (Avsec et al., 2021). More recently, ALPHAGENOME leveraged such predictions for virtual perturbation analyses (Avsec et al., 2025). These efforts collectively reduce experimental cost, enable retrospective analyses on existing data, and broaden the scope of multi-omic investigations, especially in disease contexts such as cancer, heart failure, and leukemia. Within this broader landscape, METHYLPROPHET demonstrates that accurate genome-wide DNAm prediction from transcriptome and sequence data is both feasible and biologically meaningful, thereby opening new directions for integrative epigenomic discovery.

## B  CORRELATION BETWEEN DNAM AND GENE EXPRESSION

Differences in DNAm can result in differences in gene expression. For example, using distinct tissues with paired bulk RNA-seq and WGBS data from ENCODE data portal, we found a negative relationship between the DNAm on each CpG and the expression of its nearest gene (Figure 5).

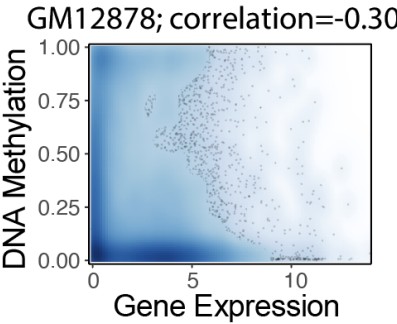

Figure 5: An example to show DNAm and the nearest gene expression is negatively correlated (Spearman correlation = -0.30).

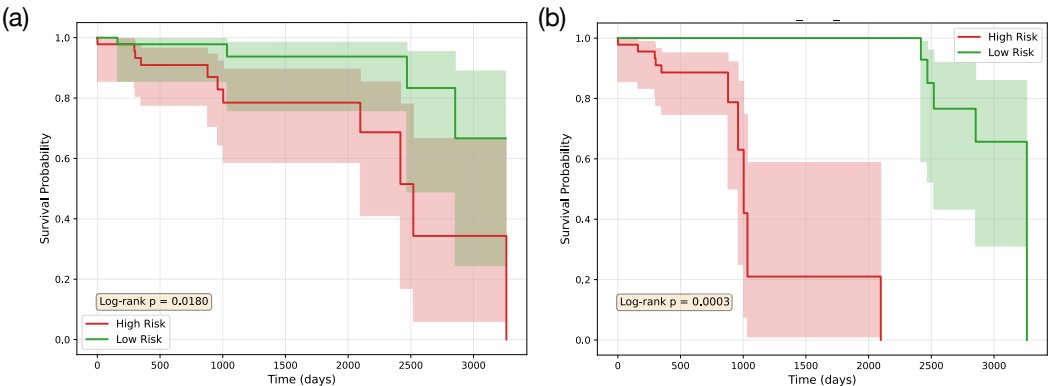

Figure 6: Kaplan–Meier survival curves for TCGA-BRCA patients based on gene-expression-only (a) and combined gene expression and DNAm signatures (b).

## C    DNA METHYLATION PROVIDES COMPLEMENTARY PROGNOSTIC INFORMATION TO GENE EXPRESSION

Although DNA methylation is known to be strongly correlated with gene expression, it remains unclear whether DNAm provides additional information beyond transcriptomic measurements alone. To directly address this question, we performed survival risk stratification analyses in TCGA-BRCA patients using gene expression alone and the combined gene expression and DNAm signatures. Patients are then stratified into high- and low-risk groups based on the median value of the risk score derived from the Cox proportional hazards model. Kaplan–Meier survival curves are generated for the two risk groups, and assessed statistical significance using the log-rank test.

As shown in Fig. 6, compared with gene expression alone (log-rank p = 0.018), the combined gene expression and DNAm signature led to a more pronounced separation between the high- and low-risk groups, with substantially increased statistical significance (log-rank p = 0.0003), indicating enhanced survival risk stratification after incorporating DNAm information.

## D    DATA

### D.1    DATA SOURCE

**ENCODE data**. Processed RNA-seq (TPM) and WGBS ($\beta$ values) data were downloaded from The Encyclopedia of Elements (ENCODE) portal (https://www.encodeproject.org/). We identified wild-type samples with both RNA-seq and WGBS profiles, along with matched summary information including species, sex, age, tissue, and bioSample information. Technical replicates were combined by averaging their gene expression and their DNA methylation profiles. The averaged

TPM values were $\log_2$-transformed after adding a pseudocount of 1. For WGBS data aligned to the hg19 genome, genome coordinates were converted to hg38 using liftover. Samples with WGBS data covering more than 80% of all CpG sites on autosomes and chromosome X were retained. Finally, all CpGs located on chromosomes X and Y were removed. A total of 95 samples covering 28,301,739 CpG sites and 55,503 genes were included in the final dataset.

**TCGA data**. Processed RNA-seq (TPM), 450K array and EPIC ($\beta$ values) data were downloaded from the Cancer Genome Atlas Program (TCGA) data portal (`https://portal.gdc.cancer.gov/`). Processed whole-genome bisulfite sequencing (WGBS) data ($\beta$ values) were downloaded from a static website provided by TCGA (`https://zwdzwd.s3.amazonaws.com/directory_listing/trackHubs_TCGA_WGBS_hg38.html`). For RNA-seq data, the TPM values were averaged for samples belonging to the same case. The averaged TPM values were $\log_2$-transformed after adding a pseudocount of 1. For 450K array and EPIC data, CpG sites with missing values across all samples were filtered out, and the $\beta$ values were averaged for samples belonging to the same case. The WGBS data provided $\beta$ values for each case. CpG sites with missing values across all cases were filtered out, and those located on chromosomes X and Y were removed. The final dataset included 9,194 450K array samples covering 408,399 CpG sites, 1,706 EPIC samples covering 740,296 CpG sites, and 32 WGBS samples covering 23,047,052 CpG sites. Additionally, gene expression profiles spanning 60,660 genes were included for each sample.

## D.2 DATA PARTITION AND PROTOCOLS

Our model takes CpG-related information and gene expressions as inputs and predicts the methylation level for the given CpG site. Originally, there are three raw files to be processed, a raw methylation beta file, a sample gene profile, and a reference human DNA sequence template (hg38. The raw methylation beta profile consists of a matrix $M \times N$, where there are $M$ CpG sites and $N$ samples, while the gene expression profile includes the expression of $L$ genes for all samples $N$. The data partition pipeline is shown in Figure 7.

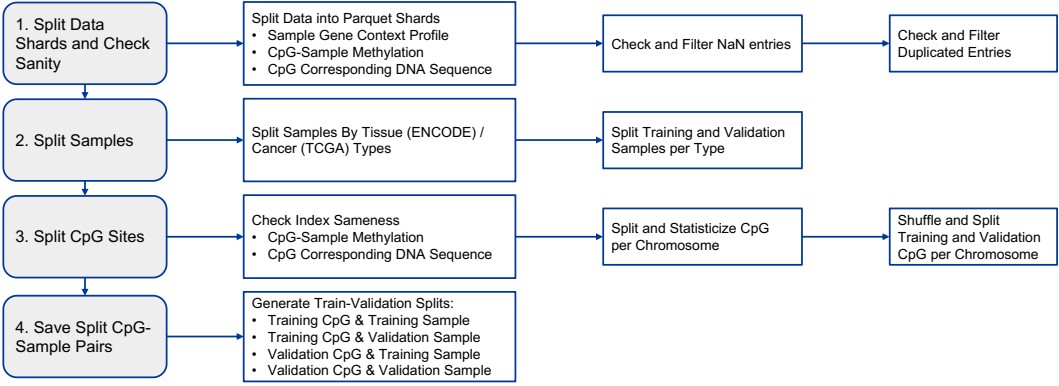

Figure 7: Data partition diagram.

**Data sharding and sanity check**. Since the raw methylation beta matrix is enormous, reaching an order of magnitude of billion (2.8 billion for ENCODE WGBS and 3 billion for TCGA Array), we first split the gigantic matrix into small shards. Sharding can leverage parallel computation and thus speed up the data pre-processing. We split the methylation beta matrix by rows (*i.e.*, by CpG sites) where every 10k rows assemble a shard file. During methylation matrix sharding, the corresponding DNA sequence for each CpG site in a shard is saved simultaneously using the reference human DNA sequence template (hg38). The window size of DNA sequence is 1Kb for the given CpG site. In addition, we filter out NaN entries and deduplicate genes and CpG sites.

**Sample split**. To split samples in to training and validation set, we first count the number of samples for each tissue / cancer types (ENCODE WGBS, Figure 8; TCGA Array, Figure 9). Then we split the samples based on the types.

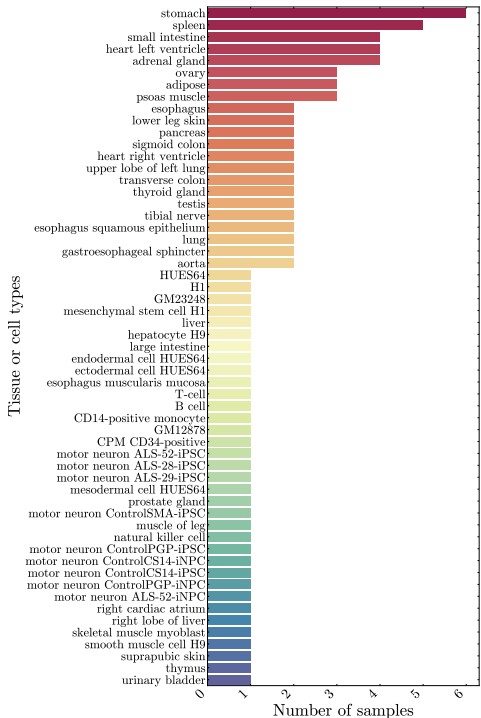

Figure 8: Samples counts by tissue types in ENCODE data.

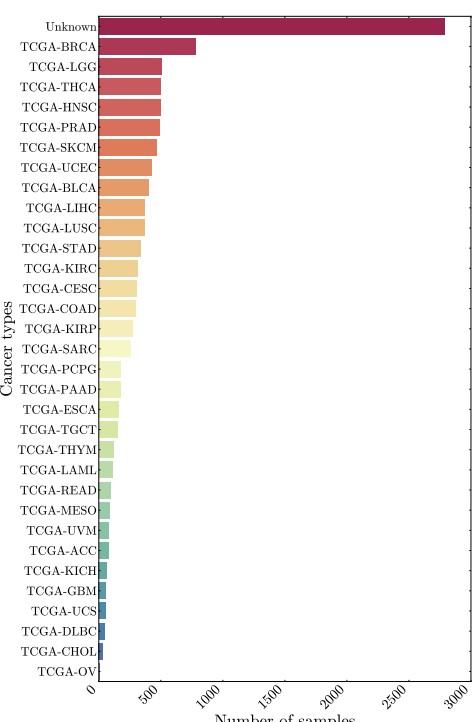

Figure 9: Samples counts by cancer types in TCGA data.

There are 57 tissue types and 95 samples in total in ENCODE. For those tissues with more than one samples, We randomly sampled half of them as the validation samples. All the rest samples are used for the training set.

In TCGA, there are 33 cancer types with 9194 samples summed up. We randomly choose 10% of the samples for each tissue type as validation samples, and the rest are left for training. For those do not have cancer type assigned, we treat them as type "Unknown".

**CpG split**. We first check the methylaton matrix and the corresponding DNA sequence have the same CpG index. Then we statisticize CpG sites for each chromosome. We randomly pick 10% for CpG sites in each chromosome as training CpG sites. For ENCODE, we temporarily sample another 10% as training split. While for TCGA, we use the rest 90% as training. We supplement TCGA with addition EPIC and WGBS data which have no intersected with Array data.

**CpG-sample split**. The CpG sample splits are based on the previous sample and CpG splits. For ENCODE WGBS and TCGA Array, we would have four splits, where the first split is used for training, and the rest three splits are used for validation and performance report:

1. "Training CpG and Training Sample", for training;

2. "Training CpG and Validation Sample", for validation;

3. "Validation CpG and Training Sample", for validation;

4. "Validation CpG and Validation Sample", for validation.

To further synergy the limited CpG sites in TCGA array data, we additionally incorporate TCGA EPIC and TCGA WGBS data, which have no intersections with TCGA array data.

### D.3 DATA PRE-PROCESSING

**CpG-specific DNA sequence**. We extract the DNA sequence around the CpG site to represent the CpG site. The window size is 1Kb for each site. Besides, we record the CpG island index, as well as

its region types (CpG island, CpG shore, CpG shelf, and CpG ocean). For those sites in CpG ocean, we assign $-1$ as their CpG island index. We embed the above information numerically.

**Gene expression**. The RNA counts are $\log_2$-transformed after adding a pseudocount of 1. Genes with mean and standard deviation below the specified cutoffs (ENCODE: mean $= 0.1$, std $= 0.1$; TCGA: mean $= 0.5$, std $= 0.5$) are filtered out. Mitochondrial, proline-rich and ribosomal protein genes are removed. As a result, 24,337 genes are retained in the ENCODE dataset and 25,017 genes in the TCGA dataset. Note that both protein-coding and non-protein-coding genes are included prior to filtering. To mitigate batch effects, we apply the quantization technique (Cui et al., 2024b) where the $\log_2$-transformed RNA counts are quantized based on their probability densities. The quantized values are then linearly mapped to the range $[0, 1]$ to mitigate batch effects. The resulting gene expression vectors are subsequently encoded in the downstream model.

## E  IMPLEMENTATION DETAILS

### E.1  CONFIGURATIONS OF METHYLPROPHET

The implementation details is shown in Table 12. For the experiments on ENCODE WGBS and TCGA chromosome 1, 2, and 3, we use 64 GPUs with 512 batch size per accelerator, taking about 1 GPU day for each experiment. While for those on TCGA chromosome 1, we use 32 GPUs with batch size 256, taking about half of GPU day for each experiment. We turn on gradient checkpointing to reduce memory usage and enable flah-attention 2 to speed up attention operator. The parameters specification and their computational cost are shown in Table 13.

Table 12: The implementation details.

| Optimization | |
|---|---|
| Optimizer | AdamW $(0.9, 0.95)$ |
| LR | 1.00E-04 |
| LR Decay Ratio | 10x |
| LR Decay | cosine |
| Weight Decay | 1.00E-03 |
| LR Warmup | Linear |
| Warmup steps | 2000 |
| Gradient Clipping | 1 |
| Data Epoch | 1 |
| Batch Size* | 256/512 |
| Accelerator Type | NVIDIA L40s |
| # Accelerator | 32/64 |
| Training Precision | Mixed bf16 |

Table 13: Model size and computation. *: Number of parameters includes the DNA tokenizer embeddings. †: FLOPs are estiamted with batch size equal 1.

| Transformer Size | # of Hidden Layers | Hidden Size | # of Attention Heads | # of Params * | FLOPs † |
|---|---|---|---|---|---|
| Base | 12 | 768 | 12 | 110M | 104G |

| MLP Size | # of Hidden Layers | Hidden Size | Bottleneck Factor | # of Params | FLOPs |
|---|---|---|---|---|---|
| B_6-Wi_1024 | 6 | 1024 | 4 | 70M | 70M |

### E.2  BASELINES

#### E.2.1  LEVY-JURGENSON ET AL. (2019B)

We implement the model described in Levy-Jurgenson et al. (2019b), which uses a multi-branch architecture with four subnetworks: two convolutional neural network (CNN) branches that process

DNA sequences around CpG sites, and two attention-based MLP branches that incorporate gene expression and CpG-gene distance, respectively. The outputs of all branches are concatenated and passed through a final regression head to predict DNAm levels. We use the original model structure as described in the paper. To ensure fairness, we apply the same input preprocessing and trained on the same data splits as MethylProphet. Our reimplementation is based on the open-source code available at: `https://github.com/YakhiniGroup/Methylation`.

### E.2.2 CpGPT (De Lima Camillo et al., 2024)

CpGPT is an imputation-based Transformer model trained via masked modeling on large-scale CpG methylation data. It learns context-aware representations of CpG sites by predicting masked methylation values based on the surrounding sequence. In our evaluation, we use the trained CpGPT-100M model to extract sample-level embeddings for 20 randomly selected samples from the Train Sample set. These embeddings are then used to predict DNAm levels at the corresponding Val CpG sites for each selected sample, following the Val CpG – Train Sample evaluation split. We use the publicly released trained model and inference code from: `https://github.com/lcamillo/CpGPT`.

## F Additional Evaluation Metrics

To complement the main performance metrics, we provide more evaluations to better understand model behavior, particularly in capturing biologically meaningful DNA methylation (DNAm) signals.

### F.1 Across-Sample PCC by DNAm Variability

We stratify CpG sites into bins according to their inter-sample DNAm variability, computed as the standard deviation of beta values across samples. For each bin, we compute the distribution of across-sample PCCs between predicted and measured methylation levels.

### F.2 PCC of DNAm Cell-Type and Tissue Differences

To assess the preservation of biological variation, we compare pairwise differences in average methylation levels between tissues or cell types, calculated for predicted and measured data. For each tissue or cell-type pair, we compute the PCC between predicted and measured methylation differences across CpG sites. High correlations indicate that the model captures inter-tissue and inter-cell-type epigenetic distinctions.

### F.3 DMR Overlapping Proportion between Measured and Predicted Values

We identify Differentially Methylated Regions (DMRs) from both measured and predicted methylation matrices using the `limma` R package. We rank DMRs by statistical significance and compute the overlap proportion between top-ranked regions from the predicted and measured DNAm matrices, across varying thresholds (e.g., top 1000, 2000 DMRs).

## G Evaluation Results

### G.1 Robustness to Missing Context DNAm

To assess the reliance on surrounding DNAm context, we conducted an ablation study by progressively reducing the percentage of available context CpG values for CpGPT. Table 14 and Table 15 report the performance across 200 held-out test samples.

When no surrounding context DNAm is available, CpGPT and MethylGPT degenerate (their output variance collapses), and the PCC metric becomes undefined. In contrast, *MethylProphet* remains stable across all levels of context sparsity due to its reliance on gene expression and DNA sequence inputs, which are independent of neighboring CpG methylation measurements.

Table 14: MAS-PCC (median across samples) under different levels of available context CpGs.

| % surrounding DNAm | CpGPT | MethylGPT | MethylProphet |
|---|---|---|---|
| 100% | 0.19 | 0.23 | 0.31 |
| 80% | 0.21 | 0.18 | 0.31 |
| 60% | 0.13 | 0.15 | 0.31 |
| 40% | 0.09 | 0.12 | 0.31 |
| 20% | 0.06 | 0.08 | 0.31 |

Table 15: MAC-PCC (median across CpGs) under different levels of available context CpGs.

| % surrounding DNAm | CpGPT | MethylGPT | MethylProphet |
|---|---|---|---|
| 100% | 0.84 | 0.78 | 0.88 |
| 80% | 0.88 | 0.69 | 0.88 |
| 60% | 0.79 | 0.63 | 0.88 |
| 40% | 0.69 | 0.54 | 0.88 |
| 20% | 0.60 | 0.49 | 0.88 |

These results highlight that *MethylProphet* is not only competitive in predictive accuracy but also substantially more robust and generalizable in low-data or missing-data settings. This robustness is especially valuable for real-world applications where measured DNAm data may be sparse or unavailable.

## G.2 METHYLPROPHET PERFORMANCE ON TCGA DATA

Figure 10 and Figure 11 illustrate the distribution of PCC for our ablation studies : 1) the effect of mixing TCGA data with different sequencing techniques. 2) the effect of increasing data scale of TCGA.

Both across-CpG PCC (Figure 12 (a, b)) and across-sample PCC (Figure 12 (c, d)) reach the highest values in the *Train CpG - Val Sample* split, indicating that the model effectively captures site-wise DNAm patterns while generalizing well to new samples. Specifically, the predictions are consistently more accurate when generalizing to new samples rather than to new CpGs compared with splits of Val CpG - Train Sample and Val CpG - Val Sample (Figure 12 (b)). If a sample exhibits high across-CpG PCC, it suggests that the within-sample variability of CpGs is well captured (Figure 12 (a)). This result is expected, as the overall DNAm profile of a sample consists of a long vector of CpG elements, and global trends in DNAm are typically easier to learn and predict. For across-sample PCC (Figure 12 (d)), we observe a large variability, particularly when generalizing to both unseen CpGs and samples. The CpGs with high across-sample PCC indicate that the model can predict the CpG's variability across samples (Figure 12 (c)) well. This is very important because the ability to predict a CpG's behavior across individuals is highly related to its potential as a therapeutic target.

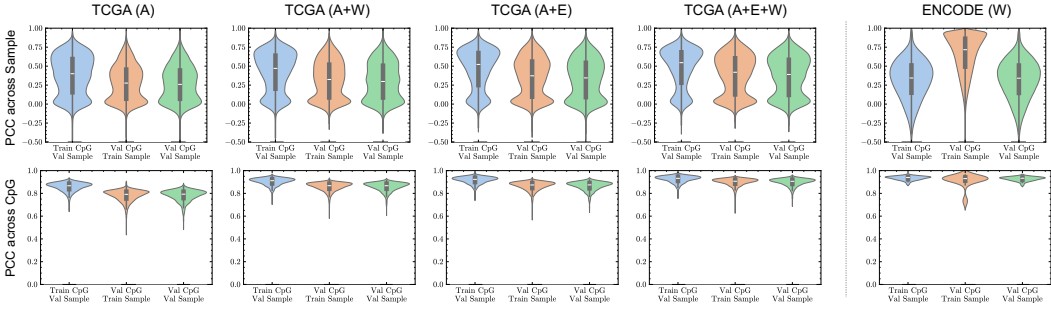

Figure 10: The distribution of PCC across Sample / CpG on validation sets for TCGA chromosome 1 data.

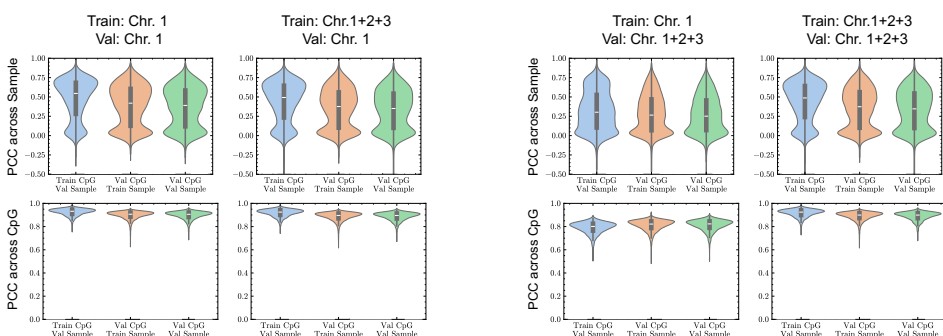

Figure 11: The distribution of PCC across Sample / CpG when increasing TCGA data scale by adding more chromosomes.

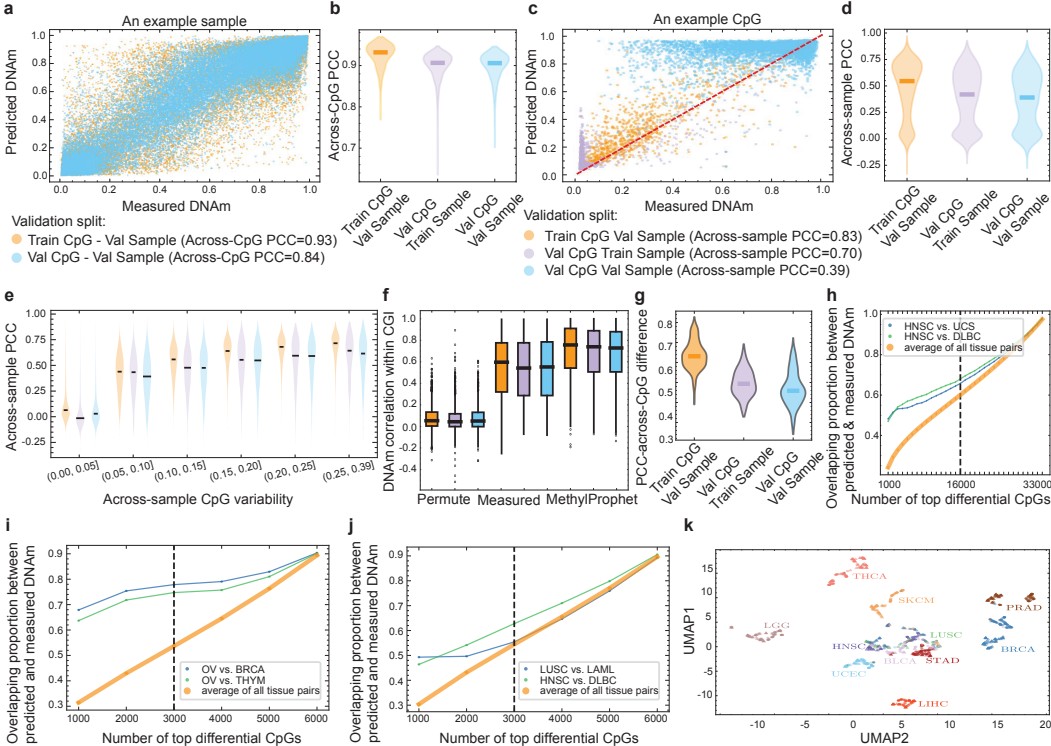

Figure 12: Cross-validation results on TCGA chromosome 1 data. (a) An example sample to demonstrate the calculation of across-CpG PCC. (b) Across-CpG PCC in three validation splits. (c) An example CpG to demonstrate the calculation of across-sample PCC. (d) Across-sample PCC in validation splits. (e) Across-sample PCC by DNAm variability in different train/validation splits, including Train CpG - Val Sample , Val CpG - Train Sample , Val CpG - Val Sample . (f) Predicted signal similarity within CGIs, with the same color scheme as (e). (g) The PCC of DNAm cell-type differences obtained from predicted and measured values. (h-j) DMR overlapping proportion between measured and predicted values. (k) UMAP of measured (triangles) and predicted (circles) samples.

We found that the across-sample PCC positively correlates with a CpG's variability across samples (Figure 12 (e)). Specifically, the highest median PCC values are observed for CpGs with a standard deviation (SD) in the range (0.25, 0.36], reaching 0.70 for Train-CpG Val-Sample, 0.63 for Val-CpG Train-Sample, and 0.60 for Val-CpG Val-Sample.

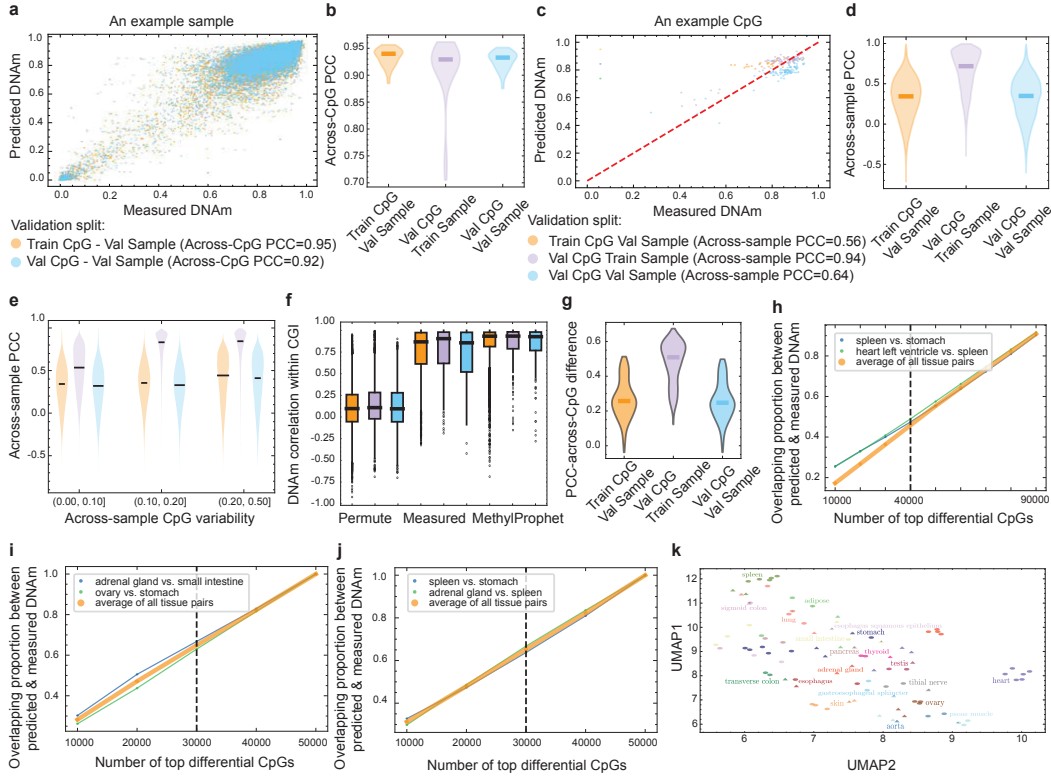

Figure 13: Cross-validation on ENCODE data. Similar to that of Figure 12, except that the results are based on the validation on ENCODE data. The sample differences (g) were calculated by comparing tissue/cell types rather than cancer types.

MethylProphet successfully maintains intra-CGI correlation patterns across different validation splits (Figure 12 (f)), indicating regional epigenetic regulation.

In addition, MethylProphet is able to preserve cancer-specific DNAm differences (Figure 12 (g)). The Train CpG - Val Sample split exhibits the highest median PCC difference, indicating that the model effectively maintains cancer-specific DNAm patterns when predicting new samples using a fixed set of CpGs. However, the Val CpG - Train Sample and Val CpG - Val Sample splits show a decline in PCC differences, suggesting reduced performance in capturing cancer-type variation when generalizing to unseen CpGs.

The differential CpGs achieves the highest overlap between predicted and measured DNAm in the Train CpG - Val Sample split, followed by Val CpG - Train Sample and Val CpG - Val Sample splits (Figure 12 (h-j)). In addition, MethylProphet-predicted DNAm landscape successfully preserves cancer-specific differences, as samples from the same cancer type remain well-clustered (Figure 12 (k)).

### G.3 METHYLPROPHET PERFORMANCE ON ENCODE DATA

Unlike TCGA, where MethylProphet performs best in the Train CpG - Val Sample split, ENCODE shows a different trend across validation splits. For across-CpG PCC (Figure 13 (a, b)), the performance is similar across splits, while for across-sample PCC (Figure 13 (c, d)), MethylProphet performs best in the *Val CpG - Train Sample* split, possibly due to the limited testing samples in EN-CODE data. Similar to that in TCGA, MethylProphet predicts methylation patterns more accurately for highly variable CpGs, where across-sample PCC increases with CpG variability (Figure 13 (e)).

In this normal tissue cohort, MethylProphet also effectively captures CpG co-methylation dynamics within CGIs (Figure 13 (f)). In the assessment of MethylProphet's ability to preserve tissue-specific DNAm differences, the Val CpG - Train Sample split exhibits the highest median PCC-across-CpG

difference (Figure 13 (g)). This contrasts with TCGA, where the Train CpG - Val Sample split performed best.

The top-ranked DMRs obtained using predicted and measured DNAm achieve a relatively high overlap across all validation splits (Figure 13 (h-j)). However, unlike in TCGA, MethylProphet performs comparably across splits. This suggests that the DMR list is more stable, likely due to the significantly larger number of CpGs included in ENCODE data. Overall, MethylProphet successfully preserves tissue differences (Figure 13 (k)), with predicted and measured samples of the same cancer types cluster together.

## H  DISCUSSION

### H.1  LIMITATIONS AND FUTURE WORKS

This work should be regarded as a proof-of-concept study that demonstrates the feasibility of leveraging gene expression and genomic context for whole-genome DNA methylation inference. While MethylProphet introduces a new paradigm and achieves promising results, we do not propose fundamentally new model architectures nor do we systematically explore more efficient or specialized designs. Instead, our focus is on establishing baseline feasibility and potential, rather than optimizing for computational efficiency or architectural innovation. Future research could address these aspects by adopting alternative architectures or scaling strategies to further improve performance and resource efficiency.

### H.2  BROAD IMPACT

While our primary objective is to enhance epigenetic research and precision medicine capabilities, we acknowledge that advances in genomic prediction technologies may have broader societal implications, including privacy considerations and ethical questions regarding genetic information accessibility. We have focused on developing methods that maintain scientific rigor while adhering to established ethical guidelines in computational biology and medical research. Our model, data source, data processing pipelines, and evaluation protocols are designed with transparency and reproducibility in mind, and we will release all code, data, protocols, and models to facilitate open scientific discourse and validation.

## I  THE USE OF LARGE LANGUAGE MODELS (LLMS)

During the preparation of this manuscript, we used OpenAI's GPT-5 model for minor language refinement and smoothing of the writing. The AI tool was not used for generating original content, conducting data analysis, or formulating core scientific ideas. All conceptual development, experimentation, and interpretation were conducted independently without reliance on AI tools.

## J  REPRODUCIBILITY STATEMENT

We full-stack release our data, code, protocols, and models in the pursuit of open science[2].

---

[2]https://github.com/xk-huang/methylprophet

