# OpenReview forum: "A New Paradigm for Genome-wide DNA Methylation Prediction Without Methylation Input"
_ICLR.cc/2026/Conference — ICLR 2026 Poster_

### Official Review · Reviewer_W98K · 2025-10-23

**Soundness:** 2
**Presentation:** 3
**Contribution:** 2
**Rating:** 2
**Confidence:** 4

**Summary:**

The paper presents a transformer-based architecture to predict DNA methylation. The key innovation is represented by the use of "paired" gene expression profiles as sources of additional context, which enables prediction at the single CpG site even in the absence of information on methylation (but with expression measurements). The proposed approach, called MethylProphet, is tested on data from major consortia (ENCODE and TGCA) demonstrating a good level of accuracy.

**Strengths:**

- The paper is well written, the task is well motivated and the approach is described in sufficient detail as to be clear whet the authors are doing and why.
- The topic is of interest, although with caveats (see weaknesses).
- The idea of using gene expression as context is novel and certainly of relevance to the task.

**Weaknesses:**

I have many reservations about the paper, here are the main ones:
- Gene expression and methylation are well known to be correlated, so in a sense adding expression as context is a natural idea. The question is whether adding expression makes the task still worthwhile. Much of the interest in methylation stems from its mechanistic role in regulating expression, as well as its biomarker function. If you already have expression measurements, the need to know much about methylation might be questionable.
- Methylation is poorly understood in general but most CpGs are either unmethylated in islands associated with active genes, or methylated in the ocean or near repressed genes. This already suggests that knowledge of expression could provide a simple model with accuracy on a comparable scale (if in island and gene on, then 0, otherwise 1).
- The comparisons are very limited, and the Fig 5 comparison against a method that does not use expression shows a fairly limited improvement despite the extra data.

**Questions:**

- Methylation arrays often provide an aggregated reading for nearby CpGs, how do you deal with that?
- How does the model perform on special CpGs, e.g. the ones used in clocks?
- How do you integrate coverage information in your loss function?

---

> ### Author Response · Authors · 2025-12-03
> **Author Response - Part 1**
>
> Thank you for your thoughtful review. We appreciate your positive assessment of the novelty, quality, clarity, and significance of our work. The main questions concern the value of DNAm prediction given its correlation with expression, the rule-based baseline comparison, the handling of array data, performance on special CpG subsets, and the treatment of coverage. We address these points below and summarize the corresponding manuscript revisions, organizing our responses into several parts for clarity.
>
> > Gene expression and methylation are well known to be correlated, so in a sense adding expression as context is a natural idea.
> The question is whether adding expression makes the task still worthwhile. Much of the interest in methylation stems from its mechanistic role in regulating expression, as well as its biomarker function. If you already have expression measurements, the need to know much about methylation might be questionable.
>
> We agree that DNA methylation (DNAm) and gene expression are correlated. However, **correlation does not imply redundancy**. Expression is a fast, dynamic readout of current cell state, while DNAm is more stable and encodes long-term regulatory and exposure history. Specifically, we clarify this from two perspectives below.
>
> - **DNAm is directly actionable in drug design and cancer therapy**.  While vast amounts of gene expression data have been generated to investigate key biological questions, they do not capture the landscape of the **regulatory mechanisms controlling the gene expression that can lead to disease phenotypes**.  Identifying epigenomic changes in DNA methylation (DNAm) that can alter gene expression in mammals has the potential to hold unexplored targeted information for early diagnosis, prognosis, and therapy. For example, while earlier studies reported DNAm alterations in cancer, recent studies have demonstrated that DNAm alterations are present in nearly all complex diseases such as, heart, Alzheimer's, inflammatory, neurodegenerative, and  immune diseases. Thus, differentially methylated regions (DMRs) between the normal and disease groups can serve as potential DNA methylation-based biomarkers. For example, Azacitidine and Decitabine are DNA methyltransferase (DNMT) inhibitors approved by the United States Food and Drug Administration (FDA) to treat myelodysplastic syndromes. **These treatments rely on knowledge beyond gene expression**.
>
> - **Even when gene expression is available, DNAm still provides additional and non-substitutable information.**
>     - Take survival analysis as an example. We performed Kaplan–Meier survival analysis on TCGA-BRCA patients using risk signatures derived from **gene expression alone** versus those derived from the **combined gene expression and DNAm features**. Patients are divided into high- and low-risk groups according to the median Cox-derived risk score. As shown in the updated Figure A2 in the manuscript, the gene-expression-only signature yielded a moderate survival separation, whereas **incorporating DNAm information resulted in a remarkably stronger separation between the two risk groups** (log-rank **p = 0.0003**), indicating improved survival risk stratification with the inclusion of DNAm.
>
>     - Technologies such as scM&T-seq, scNMT-seq, scTrio-seq, scMT-seq, scCOOL-seq, and scNOMe-seq were explicitly developed to **jointly profile DNA methylation and gene expression in the same cell**, because **neither modality alone is sufficient to characterize regulatory state**. The introductions of these works consistently emphasize that **coupling DNAm with RNA reveals chromatin accessibility, regulatory priming, and epigenetic constraints** that are invisible from RNA alone.
>
> From these perspectives, our method should be viewed as a computational analogue of multi-omic profiling: it enables recovery of a high-resolution DNAm landscape from expression + sequence when direct methylation assays or multiomics assays are unavailable, expensive, or infeasible.

---

> ### Author Response · Authors · 2025-12-03
> **Author Response - Part 2**
>
> > Methylation is poorly understood in general but most CpGs are either unmethylated in islands associated with active genes, or methylated in the ocean or near repressed genes. This already suggests that knowledge of expression could provide a simple model with accuracy on a comparable scale (if in island and gene on, then 0, otherwise 1).
>
> **1. The suggested CGI + expression “simple rule” does not achieve accuracy comparable to MethylProphet**.
>
> To directly test the reviewer’s hypothesis (“if CpG is in an island and the gene is on → 0, else 1”), we implemented a **rule-based CpG–Gene baseline**: for each CpG, we predicted methylation = 0 if the CpG lies in a CpG island and its nearest gene’s expression is above the 0.75 cohort quantile; otherwise, methylation = 1.
>
> As shown in the tables below, across both ENCODE and TCGA chr1, this rule performs **orders of magnitude worse than MethylProphet across all three evaluation settings**. For example, on ENCODE (Val CpG–Train Sample) its MAS-PCC is only 0.08, compared to 0.72 for MethylProphet; on TCGA, its MAS-PCC is essentially 0 (~=0.01), whereas MethylProphet reaches ~0.55. MSE and MAE are also an order of magnitude larger. These results show that the heuristic **does not come close to MethylProphet’s performance**.
>
> **Evaluation on Train CpG - Val Sample**
> | Dataset | Model | MAS-PCC | MAC-PCC | MSE | MAE |
> |-|----|------|---|-|-|
> |ENCODE | CpG–Gene| 0.1410 | 0.6972 | 0.1139 | 0.2413 |
> |ENCODE| MethylProphet | **0.3436** | **0.9398** | **0.0079** | **0.0608** |
> |TCGA chr1| CpG–Gene | 0.0045 | 0.4936 | 0.3431 | 0.4564 |
> |TCGA chr1| MethylProphet | **0.5455** | **0.9320** | **0.0199** | **0.0882**|
>
> **Evaluation on Val CpG - Train Sample**
> | Train set → Evaluation set| Model | MAS-PCC | MAC-PCC | MSE | MAE |
> |-|----|------|---|-|-|
> |ENCODE | CpG–Gene|0.0821 | 0.6889 | 0.1129 | 0.2378 |
> |ENCODE| MethylProphet | **0.7165** | **0.9297** | **0.0108** | **0.0679** |
> |TCGA chr1| CpG–Gene| 0.0089 | 0.5580 | 0.4140 | 0.5145 |
> |TCGA chr1| MethylProphet | **0.4194** | **0.9065** | **0.0266** | **0.1000**|
>
> **Evaluation on Val CpG - Val Sample**
> | Train set → Evaluation set| Model | MAS-PCC | MAC-PCC | MSE | MAE |
> |-|----|------|---|-|-|
> |ENCODE | CpG–Gene| 0.1407 | 0.6973 | 0.1138 | 0.2413 |
> |ENCODE| MethylProphet | **0.3411** | **0.9330** | **0.0086** | **0.0634** |
> |TCGA chr1| CpG–Gene|0.0052 | 0.5616 | 0.4199 | 0.5198 |
> |TCGA chr1| MethylProphet | **0.3904** | **0.9059** | **0.0271** | **0.1011** |
>
>
> **2. The apparent “CGI–expression” relationship holds within a sample but does not generalize across samples**.
>
> The reviewer’s statement that “most CpGs are either unmethylated in islands near active genes or methylated near repressed genes” is **partially correct within a single sample**: if we correlate each gene’s expression with the methylation of its nearest CpG island across genes, we indeed see a median Pearson correlation around -0.4, reflecting the familiar “active gene -> unmethylated promoter, repressed gene -> methylated promoter” pattern. However, if we **fix a given gene and correlate its expression with the methylation of its nearest CGI across samples**, the correlation distribution is centered near **zero (median ≈ 0)**. Thus, **cross-gene within-sample trends do not translate into a reliable cross-sample predictive rule**, which explains why the rule-based model fails in practice.
>
> **3. Biologically, CpG methylation exhibits fine-scale structure that cannot be captured by a binary island/on–off rule**.
>
> Even within CpG islands, whole genome bisulfite sequencing reveals substantial **local heterogeneity** (e.g., partially methylated regions, transcription-factor-associated boundaries). Moreover, many functionally important CpGs lie in **shores, shelves, enhancers, and gene bodies**, where the relationship between expression and methylation is **non-monotonic or context-dependent**. Our analyses show that **MethylProphet’s largest gains occur precisely on these harder, non-trivial CpGs**, which cannot be inferred from a coarse island/ocean or on/off gene rule.
>
> **4. Conclusion: MethylProphet captures regulatory structure beyond any simple CGI + expression heuristic**.
>
> While CGI status and gene expression provide a **useful coarse prior**, they are **insufficient for accurate sample-specific DNAm reconstruction**. The strong performance gap between the rule-based baseline and MethylProphet demonstrates that our model’s accuracy arises from **learning fine-grained regulatory patterns via joint sequence–expression modeling**, rather than rediscovering a simple island/ocean heuristic.

---

> ### Author Response · Authors · 2025-12-03
> **Author Response - Part 3**
>
> > The comparisons are very limited, and the Fig 5 comparison against a method that does not use expression shows a fairly limited improvement despite the extra data.
>
> The comparison shown in Fig. 5 was performed against CpGPT on a restricted subset of ~3,000 CpG sites. To better show this, here we:
> - Scaled up the number of CpG sites to the full intersection between our evaluation CpGs and those supported by CpGPT,
> - Performed random masking of CpG sites,
> - Added an additional imputation-based baseline, MethylGPT, for a broader and more balanced comparison.
>
> The results shown below:
>
> **Comparison with CpGPT**
> |                      | CpGPT  | MethylProphet |
> |----------------------|--------|----------------|
> | MAS-PCC on TCGA chr1 | 0.4794 | 0.5453         |
> | MAC-PCC on TCGA chr1 | 0.9250 | 0.9253         |
> | MAS-PCC on ENCODE    | 0.3192 | 0.3689         |
> | MAC-PCC on ENCODE    | 0.9401 | 0.9400         |
>
> **Comparison with MethylGPT**
> |                      | MethylGPT | MethylProphet |
> |----------------------|-----------|----------------|
> | MAS-PCC on TCGA chr1 | 0.4358    | 0.5488         |
> | MAC-PCC on TCGA chr1 | 0.8357    | 0.9343         |
> | MAS-PCC on ENCODE    | 0.2964    | 0.3512         |
> | MAC-PCC on ENCODE    | 0.8953    | 0.9360         |
>
> The results show that **MethylProphet outperformed existing methods CpGPT and MethylGPT with a fairly large gap**.  In fact,  CpGPT and MethylGPT are DNAm-imputation methods which rely on complementary DNAm measurements that are not available to MethylProphet. Therefore, our novel paradigm and existing imputation methods rely on different sources of auxiliary information, neither side is strictly “information-advantaged,” but rather they operate under **fundamentally different input assumptions**.
>
> We added the above results to Table 6.
>
> > Methylation arrays often provide an aggregated reading for nearby CpGs, how do you deal with that?
>
> **[Array aggregation does not bias modeling]** The fact that a probe may contain additional CpGs doesn’t change how beta values are computed. Beta value computation is based on the DNA methylation proportion for that site (for WGBS, the fraction of methylated reads; for arrays, the probe-level methylated vs. unmethylated intensity). In arrays, when some probes might cover multiple CpGs at low probability, beta values represent a slightly broadened CpG neighborhood instead of a perfectly isolated single CpG. However, the nature of measurements results in cell-aggregated signals, which dominate the signals,  while these local CpG-neighborhood effects are **second-order** compared with the main sources of variation we focus on in our analysis.  In addition, our model explicitly uses a **1 kb background DNA sequence window** rather than just using two base pairs (C and G) for each CpG site, and that should account for the nearby CpGs.

---

> ### Author Response · Authors · 2025-12-03
> **Author Response - Part 4**
>
> > How does the model perform on special CpGs, e.g. the ones used in clocks?
>
> In this response, we conducted two targeted evaluations on special CpG subsets using the TCGA chr1 dataset:
>
> **1. Clock CpGs**.
>
> We evaluated MethylProphet on epigenetic clock CpGs, defined according to Liesenfelder, Sven, et al. [1]. These CpGs are biomarkers of biological aging and represent a highly biologically constrained and clinically relevant subset of the genome.
>
> The prediction performance (MAS-PCC) on this subset, compared with the overall CpG set, is summarized below:
>
> |              | Clock CpGs  | Overall CpGs |
> |--------------|-|--------------|
> | MAS-PCC      | 0.6329    | 0.5455       |
>
>
> **2. Breast-cancer-related CpGs (Cancer vs. Normal)**.
>
> We further evaluated MethylProphet on breast-cancer-related CpGs. These CpGs are identified as significantly differential CpGs (method: limma) between TCGA-BRCA and the matched normal tissue samples.
>
>
> The prediction performance (MAS-PCC) on differential CpGs, compared with the non-differential CpG set, is summarized below:
> |              | Differential CpGs | Non-differential CpGs |
> |--------------|-|--------------|
> | MAS-PCC      | 0.4535    | 0.1554       |
>
>
> **Reference**
>
> [1] Liesenfelder, Sven, et al. "Epigenetic editing at individual age-associated CpGs affects the genome-wide epigenetic aging landscape." Nature Aging (2025): 1-13.
>
>
> > How do you integrate coverage information in your loss function?
>
> **[Coverage handled via preprocessing, not loss reweighting]** In our model, the training target is the beta value at each CpG, which summarizes the methylation proportion for that site (for WGBS, the fraction of methylated reads; for arrays, the probe-level methylated vs. unmethylated intensity). Hence, **we do not add coverage as a separate term in the loss function**. Instead, we incorporate coverage information at the **data preprocessing stage**: CpGs with low coverage (or poor detection p-values / bead counts on arrays) are excluded before model training. After this quality-control filtering, the remaining beta values have sufficient and reliable signals as the inputs to MethylProphet training and testing.

---

### Official Review · Reviewer_WQPP · 2025-11-01

**Soundness:** 4
**Presentation:** 4
**Contribution:** 3
**Rating:** 6
**Confidence:** 4

**Summary:**

The paper introduces a Transformer-based model for predicting whole-genome DNA methylation profiles from gene expression and DNA sequence alone, without requiring measured methylation values. Trained on billion-scale ENCODE and TCGA datasets, it achieves strong performance in zero-shot methylation prediction across both unmeasured CpG sites and unseen samples.

**Strengths:**

1. This paper addresses a key limitation of imputation-based methods for DNAm prediction, allowing application to entirely new samples.

2. The paper proposes a new formulation for the DNAm prediction task, which is intuitive and well-motivated.

3. The authors conduct a comprehensive evaluation which includes multiple validation splits to test model’s generalization ability.

4. The manuscript is well-written and well-structured, supported by informative results.

**Weaknesses:**

1. The paper does not clearly explain why this task should be feasible, especially why gene expression is used as an input modality.

2. It would be better to include an analysis of which patterns of CpGs are predictable and which are not.

3. Given the two types of input data, it is hard to know the contribution from each modality. The paper should include an ablation study comparing the model to sequence-only and expression-only baselines.

**Questions:**

1. How sensitive is the model's performance to the 1kb sequence window size?

---

> ### Author Response · Authors · 2025-12-03
> **Author Response - Part 1**
>
> We thank the reviewer for the thoughtful comments and suggestions. The main concerns relate to the use of gene expression, CpG predictability, the contributions of each modality, and sequence window sensitivity. We address these points below and highlight the corresponding revisions. For clarity, we organize our responses into several parts.
>
> > The paper does not clearly explain why this task should be feasible, especially why gene expression is used as an input modality.
>
> First, for **feasibility**, this is mainly because both gene expression and DNA sequences are accessible in this task (as described in Section 1 and Section 4.1), so that we can take these sources of data easily.
>
> Regarding **why we use gene expression**, we provide both biological rationale and empirical evidence to support our approach.
>
> - **Reflecting sample identity** Gene expression encodes sample identity compactly.
>     - DNAm spans ~28M CpGs, whereas expression uses only ~20k genes—over 1,000× lower dimensionality. This provides an efficient, sample-specific summary of tissue and cellular state.
> - **Unique information it provided for task**. Most importantly, expression provides a regulatory context that the DNA sequence alone cannot capture.
>     - Many CpGs share nearly identical local motifs, yet their methylation diverges depending on whether the region is transcriptionally active, repressed, or poised. Expression disambiguates CpGs with similar sequences but different regulatory environments, supplying essential context missing from sequence-only models.
> - **Prediction power due to the correlation**. Genomic data show that DNAm and expression are systematically coupled, so that one can inherently provide information for another.
>     - To provide direct empirical evidence, we randomly sample one individual from TCGA and ENCODE, respectively, and draw a scatter plot of promoter DNAm versus the expression of each CpG’s nearest gene (Figure A1 in the Appendix).  We observe a consistent genome-wide negative relationship (Spearman correlation=-0.30), indicating that gene expression carries stable, sample-specific regulatory information predictive of local methylation states.
> - **Empirical evidence** Empirically, we also did ablation studies to confirm that expression is necessary for accurate prediction. Removing expression (Table 9, row 2):
>     - The model collapses to identical predictions across samples (zero variance; MAS-PCC not applicable);
>     - MAC-PCC drops by ~0.1, demonstrating the substantial contribution of the expression modality.
>
> Hence, we believe the above biological rationale and empirical insights can explain why we use gene expression as an important source of data.
>
> > It would be better to include an analysis of which patterns of CpGs are predictable and which are not.
>
> Thanks for the suggestion. To address this, we systematically analyzed which CpGs are more predictable and which are not, summarized below.
>
> - We find that **CpGs with higher across-sample DNA methylation (DNAm) variability are substantially more predictable**, and their predictions are also more stable across samples, whereas CpGs with low variability are inherently difficult to predict. This trend is consistently observed in both the TCGA dataset (Fig. A8 (e)) and the ENCODE dataset (Fig. A9 (e)).
>     - We stratified CpGs into variability bins and computed the median across-sample PCC within each bin. The results show a strong monotonic increase in predictive accuracy as CpG variability increases. In TCGA:
>
>     | Across-sample CpG variability | MAS-PCC on Train CpG - Val Sample |MAS-PCC on Val CpG - Train Sample |MAS-PCC on Val CpG - Val Sample |
>     |-|-|-|-|
>     | (0.00, 0.05]| 0.05| -0.02| 0.03
>     | (0.05, 0.10]| 0.46| 0.46| 0.38
>     | (0.10, 0.15]| 0.56| 0.48|0.48
>     | (0.15, 0.20]| 0.63|0.55|0.55
>     | (0.20, 0.25]| 0.66| 0.60|0.60
>     | (0.25, 0.39]| 0.72| 0.63|0.61
>
>     - In addition to higher median accuracy, **the distribution of PCCs becomes progressively more concentrated for higher-variability CpGs**, as indicated by the narrowing violin shapes in Figures A8(e) and A9(e). This demonstrates that **high-variability CpGs are not only easier to predict, but also predicted more consistently across samples**, whereas low-variability CpGs suffer from strong noise domination and unstable correlations.
>
>     - This behavior is expected from both a statistical and biological perspective:
>     *Statistically*, low-variability CpGs have near-zero across-sample variance, which makes the Pearson correlation inherently unstable and noise-dominated, even when absolute errors are small.
>     *Biologically*, high-variability CpGs are enriched in regulatory regions with dynamic epigenetic remodeling (e.g., socially variable promoters, enhancers, and cancer-associated CpGs), making them more informative and learnable by the model.

---

> ### Author Response · Authors · 2025-12-03
> **Author Response - Part 2**
>
> > Given the two types of input data, it is hard to know the contribution from each modality. The paper should include an ablation study comparing the model to sequence-only and expression-only baselines.
>
> First, we want to clarify that the DNA sequence-only baseline has already been reported in Table 9 (Row 2). We additionally report the expression-only baseline and compare both single-modality settings with the full multi-modal model, as summarized below.
> |                                   | DNA sequence-only | Expression-only | MethylProphet |
> |-----------------------------------|-------------------|-----------------|---------------|
> | MAS-PCC on Train CpG - Val Sample | N/A               | 0.05            | 0.55          |
> | MAC-PCC on Train CpG - Val Sample | 0.8539            | N/A             | 0.93          |
> | MAS-PCC on Val CpG - Train Sample | N/A               | 0.04            | 0.42          |
> | MAC-PCC on Val CpG - Train Sample | 0.8607            | N/A             | 0.91          |
> | MAS-PCC on Val CpG - Val Sample   | N/A               | 0.01            | 0.39          |
> | MAC-PCC on Val CpG - Val Sample   | 0.8607            | N/A             | 0.91          |
>
> The poor performance of the expression-only baseline is not merely empirical but structurally inevitable under our problem formulation. Our task is CpG site–specific methylation prediction, where:
> - DNA provides locus-specific information,
> - Gene expression provides sample-wise global regulatory context shared across all CpGs within a sample.
>
> When DNA is removed, all CpGs from the same sample receive identical expression inputs, leading to identical predictions across CpG sites, zero prediction variance, and hence undefined MAC-PCC (denominator = 0).
>
> Conversely, in the DNA sequence-only setting, the absence of gene expression removes all sample-specific variability, causing identical predictions across samples for each CpG and thus a zero-variance condition that renders MAS-PCC undefined.
>
> We believe these ablations can already show that why we use both modalities.
>
> > How sensitive is the model's performance to the 1kb sequence window size?
>
> MethylProphet is **not overly sensitive to the exact choice of window size once the context reaches approximately 1 kb, and that 1 kb provides a good trade-off between performance and computational efficiency**.
>
> As shown in below tables, across all splits, performance improves when increasing the window from 200 bp to 500 bp and 1 kb, indicating that short-range flanking sequence alone is insufficient to fully capture the local regulatory context relevant to methylation. Performance then saturates at around 1 kb, with only marginal changes when further extending the window to 2 kb.
>
> **Evaluation on Train CpG - Val Sample**
>
> | Window size  | MAS-PCC | MAC-PCC |  MSE   |  MAE   |
> |-------|---------|---------|--------|--------|
> | 200b  | 0.3648  | 0.8193  | 0.0440 | 0.1449 |
> | 500b  | 0.3898  | 0.8419  | 0.0403 | 0.1389 |
> | 1000b | 0.4000  | 0.8669  | 0.0363 | 0.1216 |
> | 2000b | 0.3993  | 0.8718  | 0.0375 | 0.1294 |
>
>
> **Evaluation on Val CpG - Train Sample**
> | Window size  | MAS-PCC | MAC-PCC |  MSE   |  MAE   |
> |-------|---------|---------|--------|--------|
> | 200b  | 0.2325|0.7329	|0.0720	|0.1824|
> | 500b  | 0.2530|0.7648|0.0611|0.1637|
> | 1000b | 0.2769|0.7914|0.0555|0.1498|
> | 2000b | 0.2776|0.7832|0.0556|0.1478|
>
>
> **Evaluation on Val CpG - Val Sample**
> | Window size  | MAS-PCC | MAC-PCC |  MSE   |  MAE   |
> |-------|---------|---------|--------|--------|
> | 200b  |0.2122|0.7290|0.0672|0.1751|
> | 500b  | 0.2362|0.7446|0.0588|0.1613|
> | 1000b | 0.2597|0.7930|0.0557|0.1504|
> | 2000b | 0.2522|0.7951|0.0557|0.1520|

---

### Official Review · Reviewer_Epkb · 2025-11-01

**Soundness:** 3
**Presentation:** 3
**Contribution:** 3
**Rating:** 6
**Confidence:** 4

**Summary:**

This paper introduces MethylProphet, a proof-of-concept deep learning model that demonstrates the feasibility of predicting genome-wide DNA methylation using only gene expression and genomic context, without requiring any methylation data as input.

**Strengths:**

1. The methodological exposition is characterized by its clarity and high quality.
2. The objective of integrating multi-modal information is both well-defined and effective.

**Weaknesses:**

The primary issue with this paper is that while its core innovation is the mDNA-free prediction, it overlooks existing methods capable of predicting methylation sites from either gene expression alone or DNA sequence alone. Therefore, it is essential to establish a comparative baseline to demonstrate the synergistic advantage of fusing both modalities (gene expression and DNA sequence) over relying on a single modality.

**Questions:**

N/A

---

> ### Author Response · Authors · 2025-12-03
> **Author Response**
>
> We thank the reviewer for recognizing our core innovation and respond to the concerns below.
>
> > The primary issue with this paper is that while its core innovation is the mDNA-free prediction, it overlooks existing methods capable of predicting methylation sites from either gene expression alone or DNA sequence alone. Therefore, it is essential to establish a comparative baseline to demonstrate the synergistic advantage of fusing both modalities (gene expression and DNA sequence) over relying on a single modality.
>
> **[Uniqueness and sufficiency of existing baselines]** While DNA methylation governs gene expression and is pivotal in biological and disease mechanisms, to our knowledge, **no existing method predicts DNA methylation from gene expression alone or from DNA sequence alone**. The only comparable method is proposed by Levy-Jurgenson et al., [1], which uses both gene expression and DNA sequence, and **we have shown Methylprophet outperforms it** (Tables 4 and 5).  Interestingly, in October 2025 (after ICLR submission deadline), a preprint titled [“Inferring DNA Methylation from RNA-Seq in Renal Papillary Carcinoma Using Paired Multi Omics Data - a GenAI model”](https://www.authorea.com/doi/full/10.22541/au.175942409.95172063/v1)  highlighted the clinical interest of MethylProphet-like idea. However, we wanted to note that it employed the direct plug-in autoencoder and evaluated only on 20 patient samples with naive metrics, and achieved MAC-PCC 0.62, **much lower than our performance of 0.86** (on TCGA cancer samples). Hence, we believe we have already included the recent works for the fair comparison.
>
> **In the meantime, we do agree with you on the importance of evaluating the performance if only using gene expression or DNA sequence alone**.   Note that gene expression only provides a sample’s signature, while DNA sequence only provides a CpG’s signature. Therefore, using only one will lose either across-sample or across-CpG variability (i.e. predictions will be identical). To illustrate this empirically, we report the DNA sequence-only baseline in Table 9 (Row 2).  In this rebuttal, we further implemented the expression-only baseline on TCGA ch1 dataset and summarize the direct comparison with the full multi-modal model below:
> |                                   | DNA sequence-only | Expression-only | MethylProphet |
> |-|--|-------------|---------------|
> | MAS-PCC on Train CpG - Val Sample | N/A               | 0.05            | 0.55          |
> | MAC-PCC on Train CpG - Val Sample | 0.8539            | N/A             | 0.93          |
> | MAS-PCC on Val CpG - Train Sample | N/A               | 0.04            | 0.42          |
> | MAC-PCC on Val CpG - Train Sample | 0.8607            | N/A             | 0.91          |
> | MAS-PCC on Val CpG - Val Sample   | N/A               | 0.01            | 0.39          |
> | MAC-PCC on Val CpG - Val Sample   | 0.8607            | N/A             | 0.91          |
> These results show that:
> - DNA sequence-only achieves strong CpG-wise correlation (MAC-PCC ≈ 0.86), demonstrating that local sequence context provides substantial locus-level signal.
> - Expression-only yields near-zero sample-wise correlation (MAS-PCC ≈ 0.01–0.05) and fails to support CpG-level prediction.
> - The full MethylProphet model consistently outperforms both single-modality baselines by a large margin, confirming a strong and non-trivial **synergistic gain from modality fusion**.
>
> Importantly, the failure of expression-only and the limitations of DNA-only are not merely empirical phenomena but follow directly from the task formulation. Our task is CpG site–specific methylation prediction, where:
> - DNA provides locus-specific spatial information, and
> - gene expression provides sample-wise global regulatory context shared across all CpGs within a sample.
>
> When DNA is removed (expression-only), all CpGs from the same sample receive identical inputs, leading to identical predictions across CpG sites, zero CpG-wise variance, and thus undefined MAC-PCC.
>
> Conversely, when gene expression is removed (DNA-only), all samples for the same CpG receive identical inputs, leading to identical predictions across samples and undefined MAS-PCC.
>
> Therefore, existing single-modality methods based on **only DNA or only expression are fundamentally incapable of solving our CpG-level, cross-sample prediction task in a well-posed manner**. DNA-only captures **locus effects without regulatory state**, while expression-only captures **global regulatory state without spatial resolution**. Only their **fusion enables the model to simultaneously distinguish CpG loci and adapt to sample-specific regulatory conditions**, which is the source of the observed performance gain.
>
>
> **Reference**
>
> [1] Levy-Jurgenson, Alona, et al. "Predicting methylation from sequence and gene expression using deep learning with attention." International Conference on Algorithms for Computational Biology. Cham: Springer International Publishing, 2019.

---

### Official Review · Reviewer_SVbA · 2025-11-01

**Soundness:** 3
**Presentation:** 3
**Contribution:** 2
**Rating:** 4
**Confidence:** 4

**Summary:**

This paper presents MethylProphet, a Transformer model trained to prediction DNA methylation from gene expression, DNA sequence, and genomic annotations (such as CpG island annotation). Their approach is novel because it allows for whole genome DNA methylation inference for samples without any measure DNA methylation sites.

Their results show that their model can highly accurately predict methylation status across CpG sites in the genome for unseen samples. However, the performance for predicting methylation status at individual CpG sites across samples is only moderate, even for CpG site seen during training.

Overall, the results presented are interesting and well executed, but the paper lacks sufficient baselines and ablations to prove that their model beats simple imputation techniques.

**Strengths:**

- The paper is generally well written, and the problem and approach are clearly presented.
- The paper seeks to solve a problem that has not yet been tackled by other models in the field — whole genome methylation inference — and demonstrates that their approach is predictive.
- The authors validate their results on two independent datasets and multiple different methylation sequencing assays. Their results are quite robust across datasets and assays
- The validation splits and evaluation metrics are well designed.

**Weaknesses:**

In my opinion, the main weakness of this paper is a lack of sufficient baselines and ablations.
- There is only a comparison to one baseline “the CNN-based attention model in Levy-Jurgenson et al. (2019b)” and only on one of their two datasets. Moreover, from the details given in the appendix, it seems that MethylProphet and the baseline are not trained with the exact same information. In particular, it seems that MethylProphet is trained with additional genomic annotations that the CNN is not. For a fair comparison, MethylProphet should be trained with the exact same features/data as the baseline
- The point is taken that a main innovation of the model is that it can perform whole genome methylation inference, which cannot be done by many of the prior models in this space. However, it would still be worthwhile to see more comprehensive baselines against popular models in the field — such as CpGPT and DeepCpG — on the in-distribution generalization task. The results presented in Figure 5 are only against one baseline model and on one chromosome.
- For the MAC-PCC metric, it would be valuable to include another simple baseline of the mean methylation status across training samples
- There are minimal ablations performed on the components of the methylProphet model. In particular, CpG island context is highly indicative of methylation status, and an ablation should be performed to understand how much of the models performance can be attributed to this annotation alone.

Minor:
Some figures and methods could be updated for clarity.

    - For example, in table 4 MAS-PCC and MAC-CC are not defined (although they are defined later in the text)
    - In Figure 4, axes should be labeled with “samples” and “CpGs”. In addition, moving figure 4 earlier in the paper would be helpful

**Questions:**

- I do not see any results showing generalization performance between datasets - was this tested?

---

> ### Author Response · Authors · 2025-12-03
> **Author Response - Part 1**
>
> We thank the reviewer for the detailed comments. The main concerns involve baseline comparisons, ablation analyses, and cross-dataset generalization. We address each point below and indicate the corresponding revisions in the manuscript. For clarity, we organize our responses into several parts.
>
> > There is only a comparison to one baseline “the CNN-based attention model in Levy-Jurgenson et al. (2019b)” and only on one of their two datasets. Moreover, from the details given in the appendix, it seems that MethylProphet and the baseline are not trained with the exact same information. In particular, it seems that MethylProphet is trained with additional genomic annotations that the CNN is not. For a fair comparison, MethylProphet should be trained with the exact same features/data as the baseline.
>
> We conducted additional evaluations to address both the dataset coverage and feature-matching concerns. All new results are included in the revision.
>
> **1. Expanded baseline evaluation to an additional dataset**.
>
> We implemented the CNN-based attention model [1] on the ENCODE dataset. Results are reported in Table 4. Across all three evaluation protocols, **MethylProphet consistently outperforms the CNN on both ENCODE and TCGA**. Results are summarized below.
>
> **Evaluation on Train CpG - Val Sample**
> | Model | MAS-PCC | MAC-PCC | MSE | MAE |
> |-|-|-|-|-|
> | Levy-Jurgenson et al. | 0.2878 | 0.8355 | 0.0182 | 0.0875 |
> | MethylProphet | **0.3436** | **0.9398** | **0.0079** | **0.0608** |
>
> **Evaluation on Val CpG - Train Sample**
> | Model | MAS-PCC | MAC-PCC | MSE | MAE |
> |-|-|-|-|-|
> | Levy-Jurgenson et al. | 0.5453 | 0.7959 | 0.0345 | 0.1250 |
> | MethylProphet | **0.7165** | **0.9297** | **0.0108** | **0.0679** |
>
> **Evaluation on Val CpG - Val Sample**
> | Model | MAS-PCC  | MAC-PCC  | MSE  | MAE  |
> |-|-|-|-|-|
> | Levy-Jurgenson et al. | 0.1930 | 0.8037 | 0.0343 | 0.1262 |
> | MethylProphet  | **0.3411** | **0.9330** | **0.0086** | **0.0634** |
>
> **2. Feature-Matched Fair Comparison**.
>
> The CNN baseline [1] uses CpG–gene distance as an input. In contrast, MethylProphet uses CGI and chromosome, but **does not use CpG–gene distance**. To avoid any feature advantage, we removed CGI and chromosome from MethylProphet, while keeping the CNN baseline unchanged.
>
> Results under this feature-matched setting are reported in Table 10 and summarized below.
>
> **Evaluation on Train CpG - Val Sample**
> | Train set → Evaluation set| Model | MAS-PCC | MAC-PCC | MSE | MAE |
> |-|-|-|-|-|-|
> |E(W) → E(W) | Levy-Jurgenson et al.| 0.2878 | 0.8355 | 0.0182 | 0.0875 |
> |E(W) → E(W) | MethylProphet w/o genomic annotation| 0.3357  | 0.9208  | 0.0083  | 0.0635 |
> |E(W) → E(W)| MethylProphet | **0.3436** | **0.9398** | **0.0079** | **0.0608** |
> |T(A+E+W) → T(A) | Levy-Jurgenson et al.| 0.2878| 0.8355| 0.0182 |0.0875 |
> |T(A+E+W) → T(A) | MethylProphet w/o genomic annotation|  0.5353 | 0.9238 | 0.0221 | 0.0896|
> |T(A+E+W) → T(A)| MethylProphet | **0.5455** | **0.9320** | **0.0199** | **0.0882**|
>
> **Evaluation on Val CpG - Train Sample**
> | Train set → Evaluation set| Model | MAS-PCC | MAC-PCC | MSE | MAE |
> |-|-|-|-|-|-|
> |E(W) → E(W)| Levy-Jurgenson et al. | 0.5453 | 0.7959 | 0.0345 | 0.1250 |
> |E(W) → E(W) | MethylProphet w/o genomic annotation| 0.6776 | 0.9196 | 0.0133 | 0.0692|
> |E(W) → E(W)| MethylProphet | **0.7165** | **0.9297** | **0.0108** | **0.0679** |
> |T(A+E+W) → T(A) | Levy-Jurgenson et al.|0.5453| 0.7959| 0.0345 |0.1250|
> |T(A+E+W) → T(A) | MethylProphet w/o genomic annotation|  0.3948 | 0.8965 | 0.0285 | 0.1460|
> |T(A+E+W) → T(A)| MethylProphet | **0.4194** | **0.9065** | **0.0266** | **0.1000**|
>
> **Evaluation on Val CpG - Val Sample**
> | Train set → Evaluation set| Model | MAS-PCC | MAC-PCC | MSE | MAE |
> |-|-|-|-|-|-|
> |E(W) → E(W)| Levy-Jurgenson et al.  | 0.1930 | 0.8037 | 0.0343 | 0.1262 |
> |E(W) → E(W) | MethylProphet w/o genomic annotation| 0.3251 | 0.9222 | 0.0090 | 0.0645 |
> |E(W) → E(W)| MethylProphet | **0.3411** | **0.9330** | **0.0086** | **0.0634** |
> |T(A+E+W) → T(A) | Levy-Jurgenson et al.|0.1930 |0.8037| 0.0343 |0.1262|
> |T(A+E+W) → T(A) | MethylProphet w/o genomic annotation|  0.3782 | 0.8948 | 0.0300 | 0.1292 |
> |T(A+E+W) → T(A)| MethylProphet | **0.3904** | **0.9059** | **0.0271** | **0.1011** |
>
> Even under this more constrained setting, **MethylProphet still outperforms the CNN baseline across all splits and both datasets**:
> - ENCODE (Train CpG–Val Sample): MAS-PCC 0.29 → 0.34, MSE reduced by **>2x**.
> - TCGA (Train CpG–Val Sample): MAS-PCC 0.26 → 0.55, MSE reduced by **>4x**.
> - On the most challenging **Val CpG–Val Sample** split, MethylProphet consistently achieves **higher MAS-PCC and MAC-PCC** and **much lower MSE and MAE** than the CNN.
>
> **Reference**
>
> [1] Levy-Jurgenson, Alona, et al. "Predicting methylation from sequence and gene expression using deep learning with attention." International Conference on Algorithms for Computational Biology. Cham: Springer International Publishing, 2019.

---

> ### Author Response · Authors · 2025-12-03
> **Author Response - Part 2**
>
> > The point is taken that a main innovation of the model is that it can perform whole genome methylation inference, which cannot be done by many of the prior models in this space. However, it would still be worthwhile to see more comprehensive baselines against popular models in the field — such as CpGPT and DeepCpG — on the in-distribution generalization task. The results presented in Figure 5 are only against one baseline model and on one chromosome.
>
> We conducted expanded and more comprehensive comparisons against imputation-based models such as  DeepCPG, CpGPT, and MethylGPT, under matched in-distribution settings.
>
> - Additional Baseline: Expanded Comparison with DeepCpG
> We trained DeepCpG on the same ENCODE and TCGA in-distribution settings used for MethylProphet. DeepCPG assumes the availability of measured DNAm in the target sample. When evaluated under our inference-only setting (i.e., no DNAm observed in the target sample), DeepCpG performs poorly in cross-sample generalization:
>
> |                      | DeepCPG  | MethylProphet |
> |----------------------|--------|----------------|
> | MAS-PCC on TCGA chr1 | -0.0080 | **0.4194**         |
> | MAC-PCC on TCGA chr1 | 0.4237| **0.9065**         |
> | MAS-PCC on ENCODE    | 0.0317| **0.7165**         |
> | MAC-PCC on ENCODE    | 0.5560 | **0.9297**         |
>
> On ENCODE, DeepCpG achieves MAS-PCC close to zero, whereas MethylProphet reaches ~0.72; similarly large gaps are observed on TCGA chr1. These results highlight the fundamental limitation of imputation-based models when no DNAm is observed in the target sample.
>
> - On the Scope of the Original Fig. 5 Comparison
>
> The original Fig. 5 comparison was designed as a proof-of-concept demonstration, and was limited to subset of ~3,000 CpG sites. To address this limitation, we scaled up the comparison to the full intersection of CpG sites supported by both CpGPT and our in-distribution evaluation set, and performed random masking under identical protocols.
>
> |                      | CpGPT  | MethylProphet |
> |----------------------|--------|----------------|
> | MAS-PCC on TCGA chr1 | 0.4794 | **0.5453**         |
> | MAC-PCC on TCGA chr1 | 0.9250 | **0.9253**         |
> | MAS-PCC on ENCODE    | 0.3192 | **0.3689**         |
> | MAC-PCC on ENCODE    | 0.9401 | **0.9400**         |
>
> - Additional Baseline: Expanded Comparison with MethylGPT
> To further strengthen the evaluation, we additionally compared against MethylGPT
>
> |                      | MethylGPT | MethylProphet |
> |----------------------|-----------|----------------|
> | MAS-PCC on TCGA chr1 | 0.4358    | **0.5488**         |
> | MAC-PCC on TCGA chr1 | 0.8357    | **0.9343**         |
> | MAS-PCC on ENCODE    | 0.2964    | **0.3512**         |
> | MAC-PCC on ENCODE    | 0.8953    | **0.9360**         |
>
> The results above further validated our effectiveness under different settings compared with these baselines.
>
> > For the MAC-PCC metric, it would be valuable to include another simple baseline of the mean methylation status across training samples
>
> We implemented the requested baseline that predicts, for each CpG, the mean methylation value across the training samples and uses this constant value for all validation samples. As expected, this baseline achieves very high MAC‑PCC (e.g., ~0.97 on ENCODE and ~0.91 on TCGA) and low MSE, because predicting a per‑CpG constant closely matches the average signal. However, by construction it produces no variation across samples, so MAS‑PCC is undefined (N/A) and sample-wise structure is entirely lost.
>
> These results, now included in the revised manuscript, clarify that MAC‑PCC alone can be misleading: a trivial per‑CpG constant predictor performs very well on MAC‑PCC but is useless for capturing inter-sample variability. This motivates our use of both MAS‑PCC and MAC‑PCC, as well as MSE/MAE, and it shows that MethylProphet’s gains in MAS‑PCC reflect genuine improvements in modeling sample-specific methylation rather than just regression to the mean.
>
> | Dataset | Method | MAS-PCC | MAC-PCC | MSE | MAE |
> |---|---|---|---|---|---|
> | E(W)->E(W) | Sample mean Baseline | N/A | 0.9686 | 0.0048 | 0.0471 |
> | E(W)->E(W) | Methylprophet | 0.3436 | 0.9398 | 0.0079 | 0.0608 |
> | T(A+E+W)->T(A) | Sample mean Baseline | N/A | 0.9139 | 0.0246 | 0.1017 |
> | T(A+E+W)->T(A) | Methylprophet | 0.5455 | 0.9320 | 0.0199 | 0.0882 |
>
> Note that N/A is caused by the absence of variations across samples for each CpG site, as we predict it based on the mean methylation level over the training set.

---

> ### Author Response · Authors · 2025-12-03
> **Author Response - Part 3**
>
> > There are minimal ablations performed on the components of the methylProphet model. In particular, CpG island context is highly indicative of methylation status, and an ablation should be performed to understand how much of the models performance can be attributed to this annotation alone.
>
> To directly address this concern, we performed ablations of CGI and chromosome identity embeddings, while keeping all other components of MethylProphet fixed. These results below are now included in the revised manuscript (Table 10).
>
> **Evaluation on Train CpG - Val Sample**
>
> | Train set → Evaluation set| Model | MAS-PCC | MAC-PCC | MSE | MAE |
> |-|----|------|---|-|-|
> |E(W) → E(W) | MethylProphet w/o CGI| 0.3357  | 0.9208  | 0.0083  | 0.0635 |
> |E(W) → E(W) | MethylProphet w/o chromosome| 0.3105 | 0.9216 | 0.0083 | 0.0685|
> |E(W) → E(W)| MethylProphet | **0.3436** | **0.9398** | **0.0079** | **0.0608** |
> |T(A+E+W, 1) → T(A, 1) | MethylProphet w/o CGI|  0.5353 | 0.9238 | 0.0221 | 0.0896|
> |T(A+E+W, 1) → T(A, 1)| MethylProphet | **0.5455** | **0.9320** | **0.0199** | **0.0882**|
> | T(A+E+W,123)->T(A,123) | MethylProphet w/o chromosome| 0.4631 | 0.9159 | 0.0272 | 0.1001|
> | T(A+E+W,123)->T(A,123) | MethylProphet|  **0.4872** | **0.9246** | **0.0224** | **0.0919**|
>
>
> **Evaluation on Val CpG - Train Sample**
> | Train set → Evaluation set| Model | MAS-PCC | MAC-PCC | MSE | MAE |
> |-|----|------|---|-|-|
> |E(W) → E(W) | MethylProphet w/o CGI| 0.6776 | 0.9196 | 0.0133 | 0.0692|
> |E(W) → E(W) | MethylProphet w/o chromosome| 0.6959 | 0.9123 | 0.0117 | 0.0681|
> |E(W) → E(W)| MethylProphet | **0.7165** | **0.9297** | **0.0108** | **0.0679** |
> |T(A+E+W) → T(A) | MethylProphet w/o CGI|  0.3948 | 0.8965 | 0.0285 | 0.1460|
> |T(A+E+W) → T(A)| MethylProphet | **0.4194** | **0.9065** | **0.0266** | **0.1000**|
> | T(A+E+W,123)->T(A,123) | MethylProphet w/o chromosome| 0.3595 | 0.8709 | 0.0354 | 0.1264 |
> | T(A+E+W,123)->T(A,123) | MethylProphet| **0.3736** | **0.8993** |** 0.0290** | **0.1027**|
>
>
> **Evaluation on Val CpG - Val Sample**
> | Train set → Evaluation set| Model | MAS-PCC | MAC-PCC | MSE | MAE |
> |-|----|------|---|-|-|
> |E(W) → E(W) | MethylProphet w/o CGI| 0.3251 | 0.9222 | 0.0090 | 0.0645 |
> |E(W) → E(W) | MethylProphet w/o chromosome| 0.3328 | 0.9202 | 0.0090 | 0.0690 |
> |E(W) → E(W)| MethylProphet | **0.3411** | **0.9330** | **0.0086** | **0.0634** |
> |T(A+E+W) → T(A) | MethylProphet w/o CGI|  0.3782 | 0.8948 | 0.0300 | 0.1292 |
> |T(A+E+W) → T(A)| MethylProphet | **0.3904** | **0.9059** | **0.0271** | **0.1011** |
> | T(A+E+W,123)->T(A,123) | MethylProphet w/o chromosome| 0.3389 | 0.8852 | 0.0350 | 0.1215 |
> | T(A+E+W,123)->T(A,123) | MethylProphet|  **0.3460 ** |  **0.8992 ** |  **0.0295 ** |  **0.1037 ** |
>
>
> - **Effect of removing CpG island (CGI) annotations**.
>     - On ENCODE, adding CGI information improves MAS‑PCC by about 0.01–0.04 and reduces MSE across all three validation splits
>     - On TCGA, adding CGI information similarly yields consistent but moderate gains in MAS‑PCC and MAC‑PCC and reduces MSE/MAE.
>     - Importantly, even without CGI, MethylProphet still substantially outperforms all baselines, including the feature-matched Levy-Jurgenson CNN and DeepCpG. This demonstrates that while CGI provides a helpful prior, it is not the primary driver of performance. The dominant contribution arises from the gene-contextual Transformer modeling that integrates expression with local sequence.
>
> - **Effect of removing chromosome identity**.
> We further conducted a parallel ablation removing the chromosome embedding on both ENCODE and multi-chromosome TCGA training (chromosomes 1–3). The chromosome embedding provides:
>     - Consistent but modest gains (MAS-PCC increases of ~0.02–0.03),
>     - Small reductions in MSE
>     - This suggests that chromosome identity contributes weak global context (e.g., baseline methylation levels and sequence composition), but is again not the dominant factor in determining performance.
>
> **Comprehensive Feature Ablation Summary**
>
> As now clarified in the revised manuscript, we report ablations for:
> - DNA sequence only,
> - Gene expression only,
> - With vs. without CpG geographical information (CGI, chromosome).
>
> We hope these further ablations can address this concern.

---

> ### Author Response · Authors · 2025-12-03
> **Author Response - Part 4**
>
> > Minor: Some figures and methods could be updated for clarity. For example, in table 4 MAS-PCC and MAC-CC are not defined (although they are defined later in the text) In Figure 4, axes should be labeled with “samples” and “CpGs”. In addition, moving figure 4 earlier in the paper would be helpful
>
> Thank you for these suggestions. We have made the changes accordingly.
>
>
> - Clarification of MAS-PCC and MAC-PCC in Table 4.
> To improve readability and avoid forward referencing, we now explicitly include the abbreviations and brief definitions of MAS-PCC and MAC-PCC in the caption of Table 4. In addition, we have moved the evaluation metrics introduction, including their names, definition, explanation, and abbreviation to Section 4.2. All performance results are presented after that. Therefore, it will be very clear to readers when seeing them in later figures and tables.
>
> - Axis Labels in Figure 4.
> We have updated Figure 4 to explicitly label the axes as “Samples” and “CpGs”, as suggested.
>
> - Reordering of Figure 4.
> Following the reviewer’s recommendation, we have moved the original Figure 4 earlier in the paper, where it now appears as Figure 3, to better align with the flow of the method description.
>
>
> > I do not see any results showing generalization performance between datasets - was this tested?
>
> We considered both cross-dataset and external dataset generalization.
>
> - **Cross-Dataset Generalization within ENCODE and TCGA**.
>     - We would like to clarify that generalization across datasets with batch effects has indeed been explicitly tested in our study.
>     - Both ENCODE and TCGA exhibit substantial heterogeneity and batch effects within each dataset. For example, TCGA contains multiple cancer projects (e.g., BRCA, LUAD, etc.) with distinct tissue types and experimental conditions, and ENCODE includes samples generated across different laboratories and protocols. In addition, the two consortia differ in experimental platforms, processing pipelines, and cohort composition.
>
> - **External-Dataset Generalization to an Unseen Study (GTEx)**.
>     - We performed an additional external validation on GTEx [1],, a dataset that is entirely unseen during training and generated using different experimental pipelines. We randomly selected 9 GTEx samples with matched RNA-seq and WGBS data, and applied a MethylProphet model pretrained on ENCODE, without any fine-tuning. Before inference, we matched GTEx CpG sites to the ENCODE Val CpG – Val Sample split to ensure identical evaluation CpG targets.
>     - The results are summarized below, which show that **MethylProphet generalizes robustly to entirely unseen datasets without retraining**.
>
> | Evaluation set | MAS-PCC | MAC-PCC | MSE | MAE |
> |---|---|---|---|---|
> | ENCODE Val CpG - Val Sample | 0.3411 | 0.9330 | 0.0086 | 0.0634 |
> | GTEx ((Unseen Study)  | 0.3250 | 0.9308 | 0.0091 | 0.0668 |
>
>
> **Reference**
>
> [1] Lonsdale, John, et al. "The genotype-tissue expression (GTEx) project." Nature genetics 45.6 (2013): 580-585.

---

### Author Response · Authors · 2025-12-03
**General Response and Thank You Letter**

Dear Reviewers, AC, SAC, and PC,

We sincerely thank you for your time and effort in maintaining such a fair and rigorous reviewing process. We are thankful for all insightful comments from the reviewers, which make our work more solid.

Our paper introduces **MethylProphet**, a biologically informed framework for **genome-wide DNA methylation (DNAm) prediction** from gene expression and DNA sequence, **without requiring measured DNAm values**. We propose a **new paradigm**, shifting from imputation-based DNAm modeling to a more generalizable and scalable inference framework. The model is designed to reflect biological regulatory structure, achieves strong **in-distribution, out-of-distribution, and cross-dataset generalization**, and supports real-world applications where methylation data are sparse, unavailable, or impractical to measure.

We are grateful for the recognition of our contributions, especially that:
- The task is well motivated (Reviewer W98K)
- Our work tackles a previously unaddressed problem in the field: whole-genome methylation inference in entirely new samples and demonstrates strong predictive performance (Reviewers SVbA, WQPP).
- Our method overcomes a key limitation of imputation-based DNAm prediction methods, enabling application to completely unseen samples (Reviewer WQPP).
- The validation splits and evaluation metrics are well designed and appropriate for rigorously assessing generalization performance (Reviewers SVbA, WQPP).
- The idea of using gene expression as contextual information for DNAm prediction is novel and highly relevant to the problem (Reviewer W98K).

We addressed all methodological and experimental concerns raised by the reviewers, including:

- **Comprehensive and fair baseline comparisons**, including new large-scale evaluations against DeepCpG, CpGPT, MethylGPT, and the CNN-based attention model proposed by Levy–Jurgenson et al., as well as simple mean and rule-based baselines;

- **Ablation studies** isolating the contributions of DNA sequence, gene expression, CpG island annotation, and chromosome identity;

- **Single-modality feasibility and identifiability analysis**, showing that DNA-only and expression-only models are structurally insufficient for CpG-specific cross-sample DNAm prediction;

- **Sensitivity to DNA window size**, demonstrating that 1,000 bp offers a favorable trade-off between convergence speed and predictive accuracy.

- **External-dataset generalization to a completely unseen study (GTEx) without fine-tuning**, showing that MethylProphet generalizes robustly across independent cohorts and experimental pipelines, indicating strong cross-study transferability;

- **Analysis of CpG predictability patterns**, demonstrating that high-variability CpGs are more predictable and more stable across samples, with both statistical and biological justification;

- **Targeted evaluations on biologically important CpG subsets**, including epigenetic clock CpGs and cancer-associated differential CpGs, demonstrating that MethylProphet achieves consistently strong predictive performance on these biologically and clinically relevant CpG subsets;

- **Biological justification for using gene expression**, clarifying that DNAm provides complementary and non-redundant regulatory and biomarker information beyond RNA alone (including survival analysis and clinical relevance).

- **Clarification of methylation array aggregation and coverage handling**, including quality-control filtering and use of beta values as training targets;

- **Presentation and clarity improvements**, with explicit metric definitions in table captions, corrected axis labels, and improved figure ordering.

All related parts are reflected in our revision. We hope the revised paper more clearly communicates our contributions and their implications. Thank you again for your time, insight, and support.

Best regards,

The Authors of Submission 17140

---

### Meta-Review · Area_Chair_n1P3 · 2026-01-07

**Summary:**

This paper proposes a method for DNA Methylation Prediction.

**Reviewer Concerns:**

Most of the reviewer concerns are on experiments, and the authors provided extensive rebuttals and results.

**Reviewer Scores:**

The scores are on the lower side, but given the extensive results provided during rebuttals, I would think they may raise their scores if allowed, given the main concerns were on experiments.

---

### Decision · Program_Chairs · 2026-01-26

Accept (Poster)